# Bioaerosol generation by raindrops on soil

Young Soo Joung[1,2], Zhifei Ge[1] & Cullen R. Buie[1]

Aerosolized microorganisms may play an important role in climate change, disease transmission, water and soil contaminants, and geographic migration of microbes. While it is known that bioaerosols are generated when bubbles break on the surface of water containing microbes, it is largely unclear how viable soil-based microbes are transferred to the atmosphere. Here we report a previously unknown mechanism by which rain disperses soil bacteria into the air. Bubbles, tens of micrometres in size, formed inside the raindrops disperse micro-droplets containing soil bacteria during raindrop impingement. A single raindrop can transfer 0.01% of bacteria on the soil surface and the bacteria can survive more than one hour after the aerosol generation process. This work further reveals that bacteria transfer by rain is highly dependent on the regional soil profile and climate conditions.

[1] Department of Mechanical Engineering, Massachusetts Institute of Technology, 77 Massachusetts Avenue, Cambridge, Massachusetts 02139, USA.
[2] Division of Mechanical Systems Engineering, Sookmyung Women's University, 100, Cheongpa-ro 47-gil, Yongsan-gu, Seoul, Republic of Korea. Correspondence and requests for materials should be addressed to Y.S.J. (email: ysjoung@sookmyung.ac.kr) or to Z.G. (email: zhifeige@mit.edu) or to C.R.B. (email: crb@mit.edu).

Bioaerosols play an important role in climate change, human health and agricultural productivity[1–5]. In the atmosphere, bioaerosols can influence on the global climate, promoting cloud formation and ice nucleation even though their fraction is relatively small compared to all the atmospheric aerosols[6–10]. On the ground, bioaerosols can change micro-biogeography faster than many other transport mechanisms[11,12]. Because bioaerosols can lead to dispersion of biological contaminants over long distances relative to terrestrial transport mechanisms[13,14], they significantly affect changes in biodiversity and ecology as well as the propagation of biological pollutants[10,15]. Furthermore, bioaerosols can be effective carriers of pathogenic organisms to plants, animals and humans, resulting in the spread of disease[4,16]. To date, aerosols generated at water/air interfaces are considered one of the main mechanisms for transferring microbes to the environment[17–20]. While it has been generally accepted that soil can serve as an intermediate home for pathogens before they transfer to their hosts, it is not clear where and how the microbes in the soil transfer from their original habitats to the atmosphere[21–25]. Considering that most microbes prefer aqueous environments to survive, it is still a mystery to explain how viable soil-based microbes spread much further and faster than would be expected through the air[1,26,27]. Futhermore, even though we know that after a rainfall there is a rapid increase of bioaerosol concentration in the air, we have not explained the wide spread of microbes with the transfer modes discovered to date[9,28,29].

In this work, aerosols represent small water droplets suspended in the air; in particular, bioaerosols are defined as aerosols containing microbes. Recently, we discovered a new mechanism of aerosol generation by raindrops hitting soil[30] (Fig. 1a–d). We demonstrated that when a raindrop hits soils, small bubbles are formed inside the raindrop and then small droplets eject when the bubbles burst at the air/raindrop interface. Depending on soil wetting-properties and raindrop impact speed, the amount of aerosols varies. Interestingly, for particular wetting conditions and impact speeds, hundreds of aerosols are generated from a single raindrop within a few microseconds (Supplementary Movie 1). On soils with wetting properties similar to sandy-clay and clay soils, most aerosols are generated when raindrops fall at velocities corresponding to light and moderate rain[30]. Furthermore, we have shown that fluorescent dyes permeated in the soil can be dispersed by aerosols[30]. Based on our findings, under actual field conditions, it was demonstrated that organic materials can be transferred though aerosols generated by raindrops[31]. Our previous work explained the physics of aerosol generation (aerosolization) from soil; however, we still need to understand if and how bacteria in soil are transferred through the aerosolization process.

Here, we illuminate a previously unexplored mechanism that transfers bacteria from soil to air through aerosols. We first develop a visualization method for characterization of aerosols containing bacteria. Using the method, we quantitatively examine the effects of bacterial surface concentration, soil composition, raindrop impact speed, and surface temperature to identify trends in bacteria transfer from soil to air. We also verify that bacteria can indeed survive after the aerosolization process. Finally, we estimate the global transfer rate of soil bacteria by rainfall.

## Results

**Visualization of aerosols containing soil bacteria.** In this work we show that soil-borne bacteria can be transferred from soil to

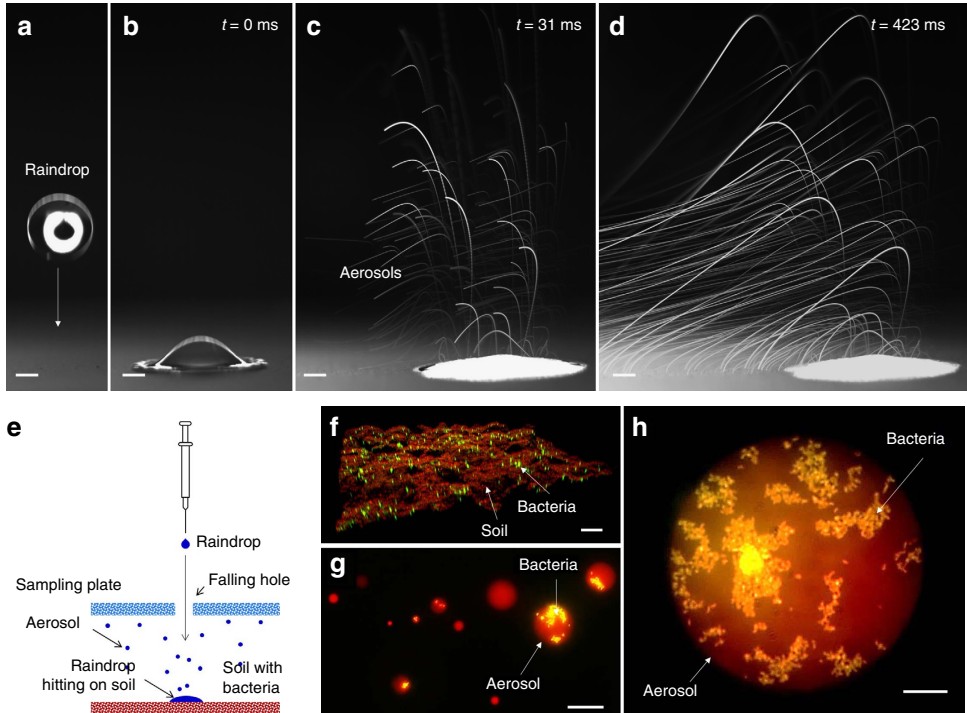

**Figure 1 | Bioaerosol generation by raindrops.** (**a**–**d**) Aerosols generated by drop impingement on a reference surface, which maximized the aerosol generation (a TLC plate (TLC-C) in Table 1). The TLC plates served as an ideal soil-like surface. The white lines are the trajectories of aerosols ejected from the initial droplet after impact over a period of 400 ms. Due to air flow above the droplet, the trajectories of the ejected aerosols are curved. The scale bars indicate 1 mm. For more details, see Supplementary Movie 1. (**e**) Schematic illustration of the experimental procedure for drop impingement on soil and aerosol collection. (**f**) Confocal microscopy images of *C. glutamicum* on the surface of clay soil with the cell density of 250 cells mm$^{-2}$. (**g,h**) Fluorescent microscopy images of aerosols generated by drop impingement on clay soil pre-permeated with *C. glutamicum*. The red circles and the yellow dots indicate aerosols and *C. glutamicum*, respectively. The scale bars indicate 200, 50 and 25 μm in **f**–**h**, respectively.

air via aerosols generated by rain. Aerosols are collected on the sampling plates positioned above the soil of interest (Fig. 1e). Fluorescent microscopy images are obtained by using soils pre-permeated with a red-fluorescence dye (Rhodamine B) and green-fluorescence (SYTO BC) bacteria (Fig. 1f). The soils were fully dried under common laboratory environment after pre-treatment with red fluorescent dye. In this work, we used three species of soil bacteria: *Corynebacterium glutamicum* (*C. glutamicum* ATCC 13032), *Bacillus subtilis* (*B. subtilis* JMA222) and *Pseudomonas syringae* (*P. syringae* ATCC 55389), and six kinds of soil: two different types of clay, sandy-clay and sand (Table 1). In addition, two different thin layer chromatography plates (TLC) were used as reference porous surfaces (ideal soil-like surfaces). The aerosols and bacteria appear as red circles (aerosols) and yellow dots (bacteria; Fig. 1g,h). Depending on the species and strains of bacteria, it is known that bacteria show different aerosolization properties[32]. In this work, we used three representative bacteria that are each abundant in soil and show long-term viability on both soils and TLC plates. These strains allow us to evaluate the effect of aerosolization on cell viability without considerable numbers of dead cells on the surfaces. In Fig. 1g, the aerosol sizes range from a few microns to hundreds of microns; but with the higher magnification microscope-lens, we can observe submicron droplets on the plates. The number of bacteria varies from zero to several thousand in a single aerosol, mainly depending on the soil types, the bacterial surface density, the surface temperature and the raindrop impact speed (Fig. 1h).

**Viability of bacteria dispersed by aerosols**. All the bacteria used in this work can be transferred from soil to air by aerosols while remaining viable. The bioaerosols are collected on the sampling plates and the plates are kept in a standard laboratory environment for a specific amount of time. Subsequently the aerosols on the plates are transferred to agar plates to measure the number of aerosols containing live bacteria. Depending upon the bacterial species and the soil, different numbers of colonies on the agar plates are observed (Fig. 2a,b). The most bioaerosols are generated on the sandy-clay soils but no bioaerosol is observed from the sand soils. This result can be explained by our previous work showing that sandy-clay soils have the optimal wetting properties for aerosolization, but sand soils absorb raindrops too fast to generate aerosols[30]. The bacteria transferred to the aerosols exhibit different viability depending on their residence time in the air ($P < 0.05$), but there is no significant difference in the soil types tested and in the bacterial species ($P > 0.05$). As a result, the three different soil bacteria remained alive even 1 h after

aerosolization. This result suggests two important conclusions; (1) bacteria can survive aerosolization by drop impingement, and (2) bacteria can be convectively transferred to distant locations while remaining viable. The viability test used in this work may not be applicable for other soil bacteria that may lose their culturability when subjected to environmental stresses or an inadequate cultivation environment. Indeed, it is well known that a small percentage of airborne bacteria can be cultured[33,34]. In fact most estimates suggest that the majority of bacteria on the planet have evaded laboratory cultivation[35–38]. In this work, we intentionally seeded culturable bacteria into the soil before aerosolization to check if the soil bacteria can survive the aerosolization process. We did not use bacteria directly recovered from environmental samples transferred by aerosols. If the aerosolization results in some bacteria remaining viable but unculturable, the number of bacteria transferred through aerosolization would be underestimated. However, this underestimation does not undermine the main result of our viability test. To minimize the limitations of viability testing, we used direct visualization of aerosols containing bacteria to count the number of bacteria transferred by aerosols; therefore, the aerosolization efficiency reported is not distorted by bacteria culturability.

**Particle dispersion by bubble bursting**. The number of bubbles formed inside a raindrop is the key factor influencing bacterial transfer (Fig. 3a and Supplementary Movie 2). With increasing bubbles formed inside a raindrop, more bacteria can be transferred by bubble bursting. The proposed mechanism is most relevant when the soil is not fully wet. After a couple of raindrops impact the same location, the soil is fully wet and a thin water film forms on the surface. In this case, we speculate that another possible mechanism of bacteria transfer by splashing is more relevant. Thoroddsen *et al.* have shown that the ejected sheet brings liquid from the bottom of the film, and when it breaks up into fine spray it could act just like the proposed bubble mechanism[39]. If we consider this mechanism, it is possible that even more bacteria can be dispersed by raindrop impact. However, in the present work, we solely deal with the dispersion of bacteria by aerosolization[39]. We investigated the effect of bubble bursting on the dispersion of particles for two different initial conditions (Fig. 3b); first, with different particle concentrations in raindrops and clean surfaces (Case 1), and second, with different particle densities on the surfaces and pure raindrops (Case 2). Interestingly, for the wide range of surface particle density and raindrop particle concentration, the number

**Table 1 | Surface properties related to bioaerosol generation by raindrops impinging on soils and TLC plates.**

| Media | Media hydraulic diffusivity | Critical surface temp. | Critical impact speed | Aerosolization efficiency: ε (%) | | | | |
|---|---|---|---|---|---|---|---|---|
| Name | $D_{cap}$ (mm$^2$ s$^{-1}$) | $T_c$ (°C) | $V_c$ (m s$^{-1}$) | 1 μm latex bead | 10 μm latex bead | *C. glutamicum* | *P. syringae* | *B. subtilis* |
| TLC$_A$ | 9.5 | 30 | 1.4 | 0.16 | 2.34 | 0.093 | NA | NA |
| TLC$_C$ | 22.4 | 30 | 1.33 | 2.24 | 2.68 | 0.16 | 0.034 | 0.0074 |
| Clay$_A$ | 2.6 | 30 | 1.53 | 0.15 | 0.87 | 0.004 | 0.0066 | 0.0021 |
| Clay$_B$ | 1.4 | 40 | 1.33 | 0.15 | 3.56 | 0.02 | 0.0392 | 0.0088 |
| Sandy clay$_A$ | 12 | 20 | 1.47 | 0.83 | NA | 0.013 | 0.0103 | 0.0034 |
| Sandy clay$_B$ | 4.8 | 50 | 1.33 | 1.24 | NA | 0.0060 | 0.0072 | 0.0102 |
| Sand$_A$ | 127.6 | 30 | 1.53 | 0.01 | 0.05 | NA | NA | NA |
| Sand$_B$ | 252.8 | NA | NA | NA | NA | NA | NA | NA |

$D_{cap}$, $T_c$ and $V_c$ are hydraulic diffusivity[54], critical temperature and critical impact velocity of the surfaces, respectively. In the case of Sand$_A$ and Sand$_B$, the dispersion of bacteria was not observed due to the low aerosolization efficiency; especially any particles and bacteria were not transferred by raindrops on Sand$_B$.

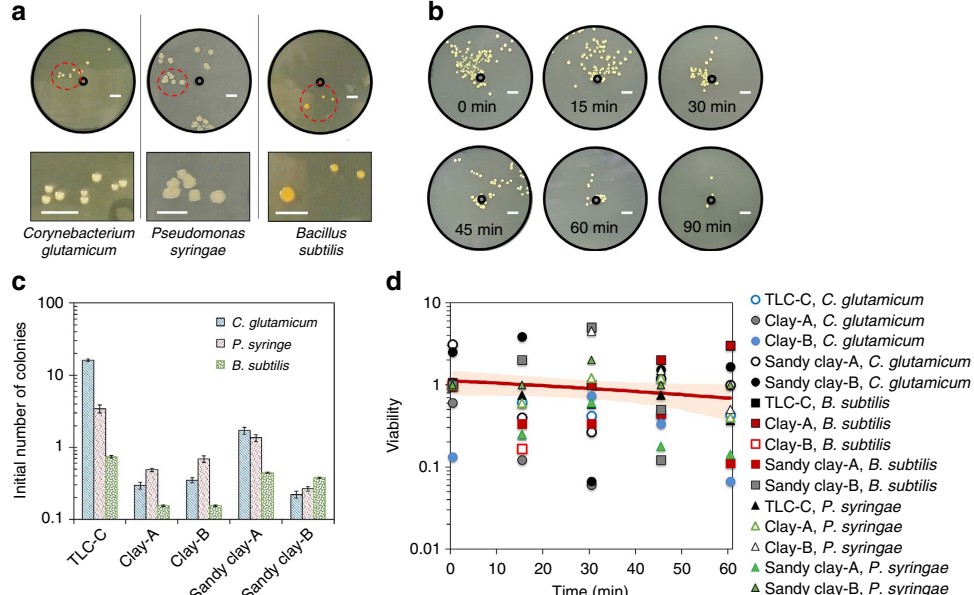

**Figure 2 | Viability of bacteria transferred by aerosols.** (**a**) Colonies of three kinds of soil bacteria, *C. glutamicum*, *P. syringae* and *B. subtilis*, cultured on agar plates for 2 days after they were aerosolized by raindrops on sandy-clay soil (Sandy clay-A in Table 1). The inner black circles indicate the location where raindrops hit on the soil. The yellow dots indicate the colonies where bacteria grew. The scale bars represent 10 mm. (**b**) Viability test with respect to the duration of drying the aerosols collected on the sampling plates. The time, displayed in the images, indicate the drying duration. Aerosols were generated from TLC plates (TLC-C in Table 1) pre-permeated with *C. glutamicum*. The colonies were cultured on agar plates for 2 days after the aerosolization. The scale bars indicate 10 mm. (**c**) Average number of colony-forming units from a single raindrop when the aerosols, collected on the sampling plates, were transferred to the agar plates immediately after aerosolization. The error bars represent ±1 s.d. resulting from nine drop impingements. The impact velocity was 1.4 m s$^{-1}$, the drop diameter of 2.8 mm, and the surface temperature 20 °C for all cases. (**d**) Viability of bacteria with respect to time after aerosolization. The viability is the ratio of the number of colonies on the agar plate to the number of aerosols containing bacteria collected on the sampling plate. For more details, see 'Methods' section.

of dispersed particles shows opposite tendencies for the two cases with respect to surface temperature (Fig. 3c). At the surface temperature of 30 °C, the least particles are dispersed for Case 1 ($P < 0.05$); however, the most particles are dispersed at 30 °C for Case 2 ($P < 0.05$). Further, the trends of both cases are opposite one another.

To further illuminate this result, we estimated the number of bubbles formed inside a droplet as a function of surface temperature. We can estimate the number of bubbles formed inside a raindrop from the aerosol size distribution (Fig. 4a). Here, when the bubbles are smaller than 1 mm, the number of aerosols exponentially decreases with decreasing bubble radius[40–43]; therefore, the number of aerosols can be expressed as $N_{aerosols} = A\exp(-3^{-1} d_{bubble})$, where $N_{aerosols}$ is the number of aerosols, $A$ is a constant value ($A \sim 7.5$) and $d_{bubble}$ is the bubble diameter[40,41]. The bubble diameter is ten times greater than the mean diameter of the aerosols[44,45]; thus, the number of aerosols can be expressed as $N_{aerosols} = N_o\exp(-30^{-1} d_{aerosol})$, where $N_o$ is a constant and $d_{aerosol}$ is the aerosol diameter. Under the assumption that the bubbles have comparable diameters, the total number of bubbles ($N_{bubbles}$) formed inside a raindrop can be estimated as $N_{bubbles} = N_o A^{-1}$; therefore, we can estimate the number of bubbles formed inside the raindrop from the aerosol size distribution.

The number of bubbles was estimated by the curve fitting method. As we expected, the number of aerosols decreases exponentially with respect to aerosol droplet size (Fig. 4a). Different constants of the exponential functions, $N_o$, are obtained for different surface temperatures. From the power law functions; we can obtain the estimated number of bubbles, $N_{bubble} = N_o A^{-1}$. The maximum number of bubbles is obtained at the surface temperature of 30–40 °C for two reference surfaces

(TLC-A and TLC-C in Table 1), respectively. When we compared the number of bubbles obtained from the curve fitting method and by counting with the digital high-speed images, we find reasonable agreement (Fig. 4b). The results obtained with two different TLC plates show that the maximum number of bubbles is formed at the surface temperature of around 30 °C. When we calculated the total volume of aerosols, the least volumes were obtained at the surface temperature of 30 °C. As a result, the total volume of aerosols governs the number of dispersed particles when the particles are dissolved in the raindrop (Case 1), but the number of bubbles mainly governs the number of dispersed particles when the particles are placed on the surface (Case 2; Fig. 4c).

**Aerosolization efficiency.** We employ aerosolization efficiency[46] ($\varepsilon$) to characterize how many particles or bacteria can be transferred by a single raindrop from soil to air. Figure 5a shows the key parameters related to aerosolization efficiency. Aerosolization efficiency is the ratio of the number of particles on the surface ($N_{particles.surface}$) to the number of particles dispersed from the surface ($N_{particles.aerosols}$) on the surface area same as the cross-sectional area of the raindrop. To obtain aerosolization efficiency, fluorescent microspheres, with average diameter of 1 µm, were placed on the surfaces with particle densities ranging from $10–10^4$ particles per mm$^2$ (Fig. 5b). After drop impingement on the surfaces, we quantified the number of microspheres in the aerosols collected on the sampling plate (Fig. 5c). To verify the mechanism of particle dispersion, we first characterized the size distribution of aerosols generated from the reference surfaces (TLC-A and TLC-B) and the soils (clay-A and sandy clay-A). The size of aerosol ranges from a few microns to a few hundred

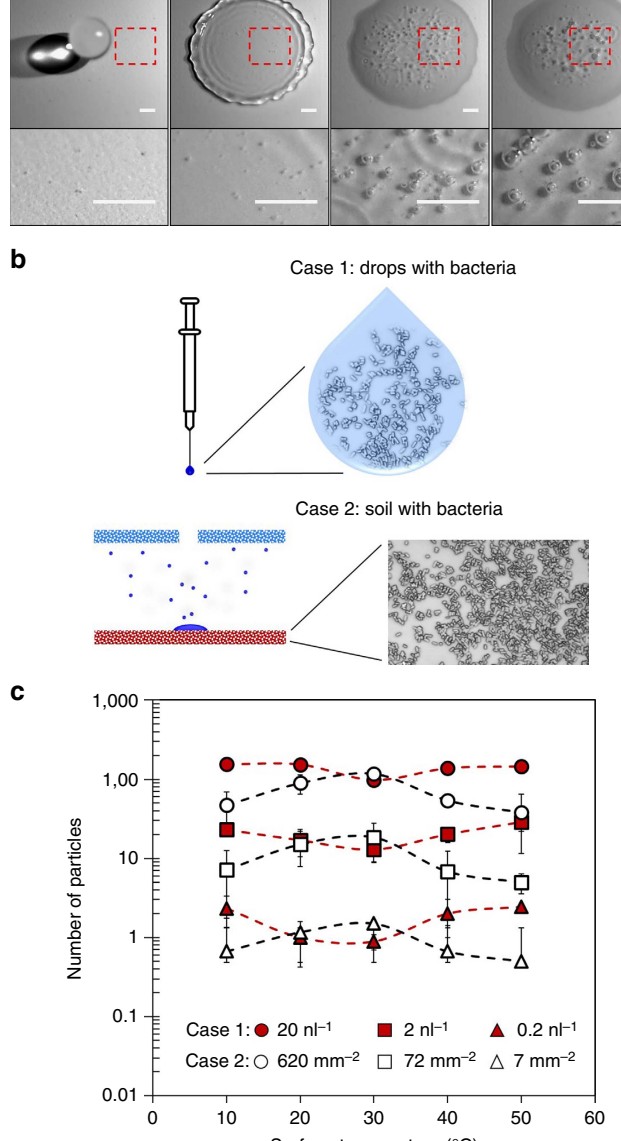

**Figure 3 | Particle transfer by bubble bursting inside raindrops.**
(**a**) Bubble formation at the interface of surface and raindrop. The red boxes indicate the regions magnified in the images below. The scale bars represent 1 mm. (**b**) Schematic illustrations of the two cases of bacteria existence. Bacteria can exist inside the raindrop (Case 1) or on the surface (Case 2). (**c**) The number of particles dispersed by aerosols generated on a TLC (TLC-C) plate with respect to surface temperature. Drop impingements were conducted with two different initial conditions: first, particles are in the raindrops (Case 1) and second, particles are on the surfaces (Case 2). In Case 1 and Case 2, different particle concentrations and densities were used; Case 1: 20 particles per nl, 2 particles per nl, and 0.2 particles per nl; Case 2: 620 particles per $mm^2$, 72 particles per $mm^2$, and 7 particles per $mm^2$. For both cases, 1 μm diameter yellow-green fluorescent microspheres were used. The red symbols and the white symbols indicate the drop impingements of Case 1 and Case 2, respectively. The error bars represent ±1 s.d. resulting from nine drop impingements.

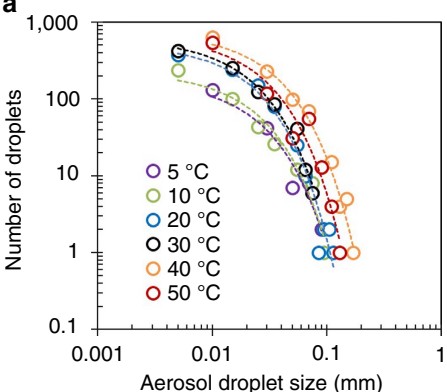

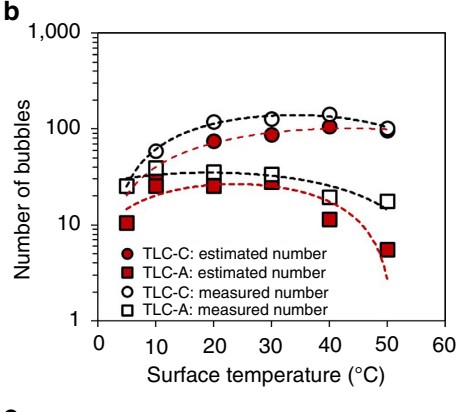

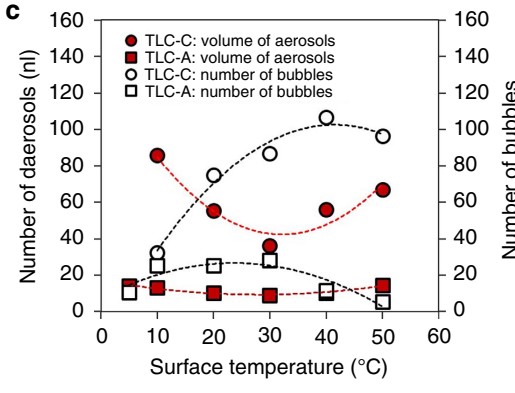

**Figure 4 | Relationship between aerosol generation and bubbles.**
(**a**) The number of aerosols as a function of aerosol diameter. From the curves, we can estimate the number of bubbles formed inside the raindrop as a function of surface temperature. The impact velocity was 1.4 m s$^{-1}$ with the raindrop diameter of 2.8 mm for all the surface temperatures. (**b**) The number of bubbles estimated by the theory (the red symbols) and counted using high-speed images (the white symbols). The theoretical data were estimated by curve fittings and an empirical equation reported[40,41]. (**c**) The number of bubbles created inside a droplet (the white symbols) and the total volume of aerosols (the red symbols) with respect to surface temperature.

microns and the number of aerosols decreases exponentially with increasing aerosol size regardless of the surfaces used (Fig. 5d). On the clay and sandy clay soils, we observed more than one hundred aerosols smaller than 10 μm from a single drop impingement. This result indicates that micro bubbles are formed

inside the drops that impact the soils and the particles are dispersed when bubbles burst at the surface of droplets.

We observe a linear relationship between the surface particle density ($S_{surface}$), which is the number of particles ($N_{particles.surface}$) per unit area and the total number of particles ($N_{particles.aerosols}$) dispersed by aerosols generated by a single raindrop (Fig. 5e). To evaluate the aerosolization efficiency of particle transfer, we introduce the dispersed particle density ($S_{aerosols}$), which can be expressed as $S_{aerosols} = N_{particles.aerosols} \times A_{raindrop}^{-1}$, where $A_{raindrop}$

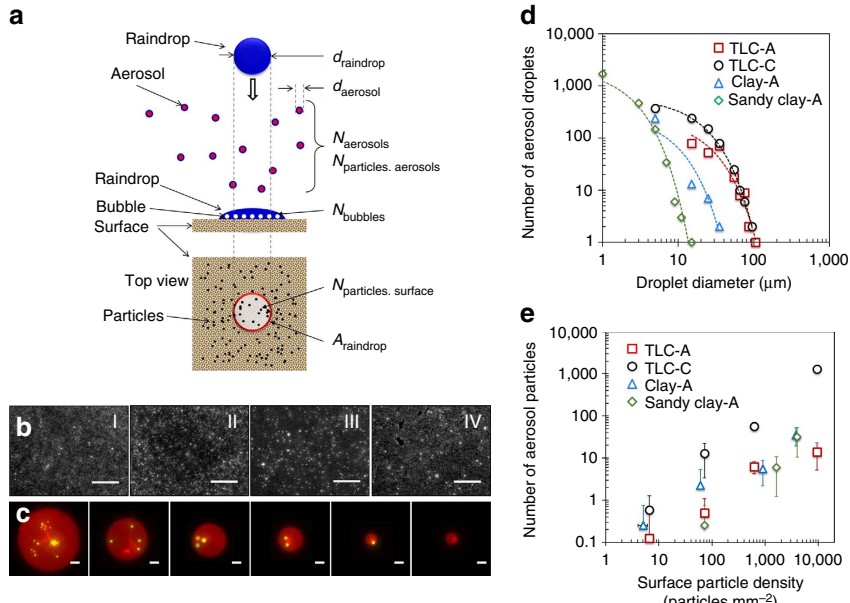

**Figure 5 | Particle transfer under different particle densities on soils. (a)** Schematic illustration of the key parameters related to particle dispersion by raindrop impact. **(b)** Fluorescence microscopy images of surfaces with different particle densities. The bright dots are 1 μm diameter yellow-green fluorescent microspheres. The approximate surface particle density of images I, II, III and IV are 10, $10^2$, $10^3$ and $10^4$ particles per mm$^2$, respectively. The scale bars represent 500 μm. **(c)** Aerosols containing 1 μm microspheres and collected on a sampling plate. The scale bars represent 25 μm. **(d)** The number of aerosols decreases exponentially with respect to aerosol diameter. **(e)** The total number of microspheres dispersed by a single raindrop is linearly proportional to the surface density of the microspheres. The surface temperature is 25 °C and the raindrop velocity at impact is 1.4 m s$^{-1}$ for **d** and **e**. The error bars indicate ± 1 s.d. resulting from nine drop impingements. The dotted lines in **d** indicate exponential fitting lines.

is the cross-sectional area of the raindrop. Since $S_{aerosols}$ also has a linear relationship with $S_{surface}$, we can express their relationship as $S_{aerosols} = \varepsilon \times S_{surface}$, where $\varepsilon$ is a constant denoted the aerosolization efficiency. The number of particles delivered by a single bubble bursting ($N_{particles.bubble}$) can be estimated as $N_{particles.bubble} = S_{aerosols} \times A_{raindrop} \times N_{bubbles}^{-1} = \varepsilon \times S_{surface} \times S_{bubble}^{-1}$, where $S_{bubble}$ is the surface bubble density defined as the number of bubbles formed in the raindrop after impingement divided by the cross-sectional area of the raindrop, $N_{bubbles} \times A_{raindrop}^{-1}$. Ultimately, the total number of particles transferred is governed by the surface particle density ($S_{surface}$), the surface bubble density ($S_{bubble}$) and the aerosolization efficiency ($\varepsilon$). These three parameters are necessary to understand how many particles are dispersed by a raindrop impacting a porous surface.

**Critical surface temperature.** Soil surface temperature plays an important role in bacterial dispersion by raindrops. We find a certain surface temperature at which the maximum particle (bacteria or abiotic particle) is transferred by aerosols from the surface (Fig. 6a). The maximum number of dispersed particles occurs at a particular temperature (in the range from 20 to 40 °C) depending on the soil type. The difference between the maximum and the minimum number of particles can be a factor of 10, indicating that surface temperature has an important role in particle dispersion from soil to air. Here, we define the surface temperature where the maximum particle dispersion occurs as the critical surface temperature $T_c$. Our experiments show that the critical temperature is not significantly affected by the size or type of particle, within a 10 °C range. For example, we obtained the same critical temperature for bacteria and microspheres. The effect of surface temperature is related to the number of bubbles formed inside the raindrop. At the critical surface temperature, the most bubbles were observed, resulting in the maximum dispersion of bacteria as shown in Fig. 4c (Case 2).

**Critical impact velocity.** Rainfall intensity is another important factor governing the bacterial dispersion by raindrops. We find that each soil has a critical impact velocity, $V_c$, where the largest number of bacteria can be transferred by drop impingement. In experiments with microspheres, with increasing impact velocity up to roughly 1.4–1.7 m s$^{-1}$, the most microspheres are transferred via aerosolization. However, the number of microspheres transferred decreases when the impact velocity is higher than the critical impact velocity (Fig. 6b). It is known that impact velocity changes the maximum radius and the minimum height of the expanding raindrop hitting soil[30]. At impact velocities less than the critical velocity, the area of soil covered by a raindrop increases with velocity, resulting in a larger surface area to form more bubbles, which causes more particles or bacteria to be transferred. However, at impact velocities higher than the critical velocity, the bubble formation is limited by the raindrop height after impact; therefore, the number of dispersed bacteria decreases due to the decreasing bubble size and number, even though the wetted surface area is increasing. As the critical surface temperature, the critical velocity is largely independent of the type of particle on the surface, yielding similar values for different size microspheres and bacteria (Table 1).

**Discussion**

To estimate the transfer rate of soil bacteria by aerosols, it is necessary to know the regional soil profile and climate conditions. Dispersed particle density is normalized by aerosolization efficiency, $\varepsilon$, obtained from the slope of the maximum dispersed particle density versus the surface particle density, as shown in Fig. 7a. The dimensionless concentration ($C^\star$) is obtained from the ratio of the dispersed particle density, $S_{aerosols}$, to the surface particle density, $S_{surface}$. The normalized particle concentration ($C_a^\star = C^\star \times \varepsilon^{-1}$) helps illustrate the influence of surface temperature and impact velocity on the particle transfer from soil to air. When the normalized particle density is plotted with respect

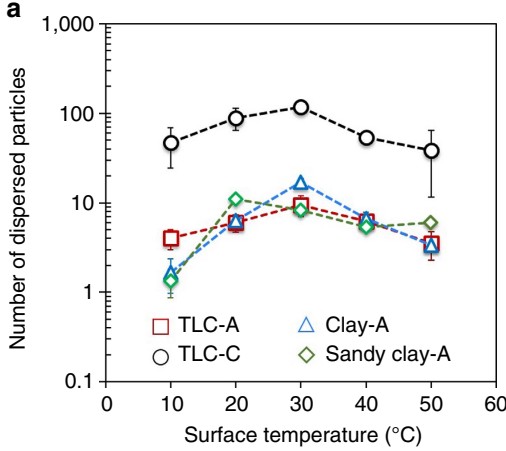

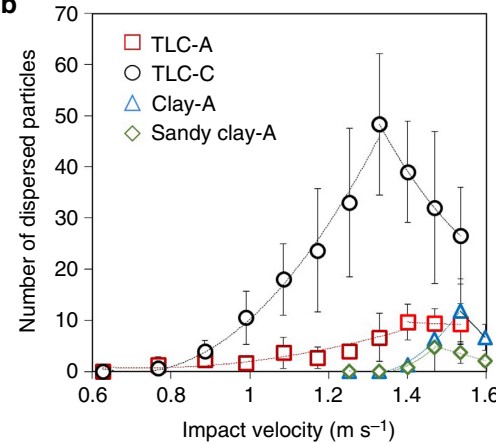

**Figure 6 | Particle transfer under different surface temperatures and impact conditions.** (**a**) The number of particles dispersed by a single drop impingement with respect to surface temperature. The surface temperature was varied from 10 to 50 °C. (**b**) The number of microspheres dispersed by aerosols at different impact velocities. Surface particle densities of TLC-A, TLC-C, and Clay-A, and Sandy clay-A are 623, 627, 1,308 and 1,725 particles per mm$^2$, respectively, in **a,b**. The raindrop velocity at impact is 1.4 m s$^{-1}$ for **a** and the surface temperature is 25 °C for **b**. The error bars indicate ± 1 s.d. resulting from nine drop impingements. The dotted lines in **b** indicate the second order polynomial fitting lines.

to the normalized impact velocity ($V^* = V \times V_c^{-1}$) and normalized surface temperature ($T^* = T \times T_c^{-1}$), we observe two peaks of maximum $C_a^*$ at $V^*$ and $T^*$ and the general trends of bacteria dispersion regarding surface and impact conditions (Fig. 7b,c). Figure 7b,c were obtained using the three kinds of soil bacteria and the two different sizes of microspheres on the six types of soils and the two different TLC plates in Table 1. As a result, we can estimate the number of bacteria that can be transferred by aerosol as a function of soil type, bacterial surface density, rainfall velocity and soil surface temperature.

The number of aerosols containing bacteria varies with soil type. Sandy clay soils showed the most bioaerosols while sandy soils do not appear to produce bioaerosols. This result is consistent with our previous work showing that more aerosols can be generated on surfaces with wetting properties similar to sandy clay soils. Sandy clay soils have ideal properties for aerosol generation and show the highest aerosol generation relative to the other soils tested, as shown in Fig. 6. Sandy soils have fast water absorption speeds such that aerosols are not generated by raindrop impact because bubbles do not form. The aerosolization efficiency of soil bacteria allows us to estimate the number of

bacteria transferred by rainfall. However, it is challenging to know the precise degree of bacteria transfer because there are diverse varieties of soil types, precipitation characteristics and environmental conditions on the planet; therefore, here we show the order of magnitude of the maximum and minimum transfer amount of bacteria based on the reported data related to bacteria surface density, soil surface area, precipitation frequency and aerosolization efficiency. Considering the aerosolization efficiency, mean bacteria surface density[9], global land area[47] and precipitation patterns[48], we estimate that the total number of bacteria dispersed by raindrops can range from $1.2 \times 10^{22}$–$8.5 \times 10^{23}$ cells per year (see the Supplementary Note 1). Others have estimated that the total number of bacteria emitted from land ranges from $1.5 \times 10^{23}$–$3.5 \times 10^{24}$ cells per year[49]. As a result, global precipitation can transfer 1.6–25% of the total amount of bacteria emitted from land. Even though, in this calculation, we did not consider other kinds of microbes such as viruses, spores and eukaryotic microorganisms as well as chemicals and solid particles, we find that a considerable population of bacteria could be dispersed by raindrops. However, this estimate should be considered an upper bound because successive raindrops can wash the aerosols out of the air. As discussed in our previous work[30], however, aerosols can be convectively migrated by wind, and thus it is expected that a reasonable fraction of bioaerosols can be dispersed into the atmosphere and evade washout by successive raindrops. Others have shown that aerosol concentration in the atmosphere rapidly increases after rainfall, suggesting that washout does not remove all aerosols[9,28,29].

In conclusion, this work has importance in three aspects; first, we have observed a new mechanism of bioaerosol generation caused by rain. Once bacteria are dispersed through aerosols, they can be transported by wind to other places much faster than other modes of transfer such as diffusion through soil. Second, we have identified experimental methods to collect and visualize aerosols including bacteria dispersed from soil. Using this method, we defined a new surface property denoted aerosolization efficiency, which describes how many bacteria can be transferred under given conditions. Third, we found that the mechanism of bacteria transfer by raindrops with bubble formation affected by surface temperature and impact speed. With understanding this mechanism, one can begin to develop methods of preventing or promoting bacterial transfer. As a result, we can predict the locations and environmental conditions that promote the transfer of bacteria from soil to air. These findings reveal a novel mechanism of bacteria transfer from soil to the environment. This mechanism can be relevant for the investigation of climate change, pathogenic disease transmission and geographic migration of bacteria.

## Methods

**Cell culture.** *C. glutamicum* (ATCC 13032) transformed to encode kanamycin resistance was cultured in brain heart infusion medium (BD, Sparks, MD, USA) with 0.5 M sucrose (Sigma-Aldrich, St Louis, MO, USA; BHIS medium) containing 50 µg ml$^{-1}$ of kanamycin at 37 °C. *B. subtilis* JMA222 was cultured in Lysogeny broth (LB) medium at 37 °C. *P. syringae* ESC 10 (ATCC 55389) was cultured at nutrient broth at 30 °C. All cells were cultured stationary phase. Optical density at 600 nm (OD$_{600}$) of the cells was measured with ultraviolet–vis spectrophotometer (UV-1800 spectrophotometer, Shimadzu Scientific Inc., Kyoto, Japan) and then diluted in BHIS medium (for *C. glutamicum*), LB medium (for *B. subtilis*) or nutrient broth medium (for *P. syringae*) to OD$_{600}$ equal to one. *C. glutamicum* was transformed with the plasmid pEC-K18mob2 using the method described by M.E. van der Rest *et al.*[50] to express kanamycin resistance as the selection marker. The transformed cells were recovered in LBHIS agar plates (per liter of LBHIS agar consists of 12.5 g LB broth, 18.5 brain heart infusion powder, 15 g agar and 91 g sorbitol) containing 50 µg ml$^{-1}$ of kanamycin. *B. subtilis* and *P. syringae* were not transformed. However, when using *B. subtilis* and *P. syringae,* negative controls were conducted to confirm that experiments were not contaminated from the environment.

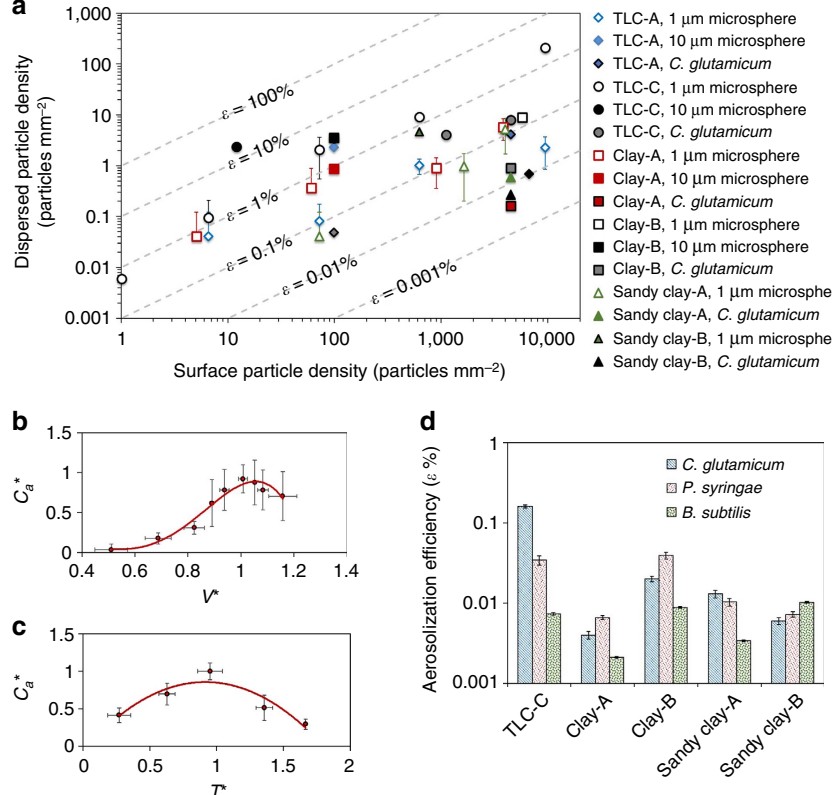

**Figure 7 | Bacterial dispersion through aerosols generated by raindrops. (a)** The maximum cell density of aerosols dispersed from the surfaces pre-permeated by microspheres or bacteria. **(b)** Normalized cell density $C_a^* = C^* \times \varepsilon^{-1}$ with respect to non-dimensional surface temperature, $T^*$, which is the ratio of the surface temperature $T$ to the critical surface temperature $T_c$, where $C^*$ is the ratio of the dispersed cell density to the surface cell density, $\varepsilon$ is the aerosolization efficiency obtained from **a**, and $T_c$ is the surface temperature at which the maximum dispersed cell density is obtained. **(c)** Normalized cell density with respect to the dimensionless impact velocity, which is the ratio of the impact velocity $V$ to the critical impact velocity $V_c$ at which the maximum dispersed cell density is obtained. For **b,c**, six types of soils, two kinds of reference surfaces, three different species of bacteria, and two different sizes of microspheres were used as shown in Table 1. The red lines in **b,c** indicate the polynomial fitting lines. The error bars represent ±1 s.d. resulting from more than 90 drop impingements. **(d)** Average aerosolization efficiency of three kinds of soil bacteria on four different soils and a reference surface (Table 1). On Sand-A and Sand-B, no bioaerosol was observed. The error bars represent ±1 s.d. resulting from nine drop impingements.

**Preparation of suspensions of microspheres.** Three different sized microspheres were purchased from Sigma-Aldrich (carboxylate-modified polystyrene, fluorescent yellow-green, 0.1 and 1 μm, and micro particles based on polystyrene, dark red, 10 μm). The microsphere suspensions were used without further purifications to make the desired concentration with 1% PBS solutions. The microsphere concentrations were estimated in terms of the number of particles per unit volume from the particle concentration information provided by the supplier.

**Preparation of surfaces pre-permeated with microsphere or bacteria.** Before treatment of soils with microspheres or bacteria, the soils were sterilized and dried for a day. To prepare soils pre-permeated with microspheres or bacteria, the microsphere or bacterial suspensions were placed on the soil surface with a specific volume density (0.05 ml cm$^{-2}$). To prepare TLC plates permeated with microspheres or bacteria, the TLC plates were soaked in the relevant suspension for 30 s and then dried for 25 min in a common laboratory environment. The surface particle or cell density was measured with digital microscopy. Using image characterization software (ImageJ (refs 51,52)), the number of microspheres or bacteria were counted and then the surface particle or cell density was calculated by the observed surface area. Six different soils were used; clay, sandy clay and sand were purchased from Sigma-Aldrich (Clean clay #5, PH–Sandy clay, and Clean sand #4,), another clay and sand from World's Science (Nasco soil samples, Fort Atkinson, WI, USA), and the other sandy clay soil was collected at Boston area. Two TLC plates were purchased from Sigma-Aldrich (silica gel and aluminium oxide matrixes).

**Visualization of aerosols containing bacteria.** To visualize cell-containing aerosol on the sampling plates, we pre-treat soils with fluorescently labelled bacterial suspension. Two fluorescent dyes were used to indicate the cells and the aerosols. SYTO BC (Life Technologies, Carlsbad, CA, USA) was used to stain bacteria with green fluorescence (excitation peak at 485 nm and emission peak at

500 nm). Final concentration of 5 μM of SYTO BC was added to the cell suspension. Rhodamine B (Sigma-Aldrich) was added to the cell suspension to visualize the aerosols with red fluorescence (excitation peak at 540 nm and emission peak at 625 nm). The final concentration of Rhodamine B was 0.001 g l$^{-1}$. Raindrops, which come from deionized water, hit porous media, generating aerosols. The aerosols were collected on the sampling plate. The aerosols and the bacteria were detected by different filters in the fluorescence microscope, because Rhodamine B was dissolved into the aerosols. In this work, we did not consider evaporation when calculating aerosol size because the time scale from aerosol generation to aerosol impact and measurement on the sampling plate is much smaller than the time scale of evaporation. The mean velocity of aerosol ejections is 10 m s$^{-1}$ and the gap distance from the surface to the sampling plate was 10 mm, so the impact time scale is roughly 1 microsecond. Considering the flight velocity and the mean aerosol size under the common environment condition of 20 °C and 25% RH, the diameter change for an aerosol with initial diameter of 100 μm is <0.04%, so the effect of evaporation on measured size is not significant[53].

**Drop impingement on soils with bacteria and microspheres.** Drop impingements with deionized water were conducted on the prepared soils and TLC plates pre-permeated with cells or microspheres. The impact velocity of raindrops was varied with different dropping heights from the surfaces. The raindrops hit on the surfaces after passing through the hole at the centre of the sampling plate located above the surface. The height of the sampling plate was varied from 2 to 15 mm to find the optimal height of the sampling plate for the maximum collection efficiency. In this work, we used 10 mm of the sampling height. The surface temperature of soils and TLC plates was varied from 5 to 50 °C with a house-made temperature controller. The impact speed varies from 0.6 to 1.7 m s$^{-1}$. The impact speed was estimated using the drop release height and calibrated with high-speed imaging. We used a high-speed camera (Photron, San Diego, CA, USA) to record the process of drop impingement, bubble formation and bubble bursting inside the

droplets. The diameter and number of bubbles were measured by an image analysis programme (ImageJ (refs 51,52)) applied to the high-speed movies.

**Cell viability.** For cell viability on the surfaces, porous substrates, including TLC plates and soil samples, were pre-permeated with cell suspension. The cell pre-permeated porous samples were dried for 25 min. Then, the porous samples stamped the kanamycin-containing LBHIS agar plates to recover *C. glutamicum*, LB agar plates to recover *B. subtilis* or nutrient agar plates to recover *P. syringae*. We found that the *C. glutamicum* can survive on the soils and TLC plates after drying. To test cell viability after aerosolization, the soils and TLC plates were permeated with bacterial suspensions and kept in a common laboratory environment for 25 min before aerosolization experiments. We diluted the cells directly starting with nutrient media. After the surfaces were permeated with the cell suspension we evaluated the aerosolization process after different amounts of drying time. We find that the original wetting properties of the surfaces were fully recovered within 10 min after permeation. To be careful, we dried the surfaces for 25 min to prevent changes in wetting properties of the soils and TLC plates during the aerosolization experiments. The sampling plates with aerosols generated from soils and TLC plates pre-permeated with bacteria were maintained in the common environment condition during different times from 0 to 90 min and then the sampling plates were stamped on the corresponding agar plates. The agar plates were incubated in a shaking incubator at 37 °C for *C. glutamicum* and *B. subtilis* and 30 °C for *P. syringae* for 2 days and colony-forming units were counted. The viability data were obtained with three different species of bacteria and five types of soil at 15 min increments for 60 min. The regression analysis was performed to identify correlation between drying time and the viability with a 95% confidence level (when $P < 0.05$). The Student's *t*-test was used to identify significant differences between soil types and bacteria species with a 95% confidence level (when $P < 0.05$).

**Data availability.** The authors confirm that all relevant data to this study are available in the article and the Supplementary Information or from the corresponding author upon request.

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

## Acknowledgements
We are grateful to Dr Jim Bales and Ms Sandra J. Lipnoski of MIT Edgerton Center for allowing us to use their high-speed camera. We thank Qianru Wang for the microscopy images (Fig. 3b) of bacteria.

## Author contributions
Y.S.J., Z.G. and C.R.B. designed the research; Y.S.J. and Z.G. performed the research; Y.S.J., Z.G. and C.R.B. analysed the data. Y.S.J., Z.G., and C.R.B. wrote the manuscript.

## Additional information

**Competing financial interests:** The authors declare no competing financial interests.

**Publisher's note**: 

