## [Peer Review file · Nature Communications]

Reviewers' Comments:

Reviewer #1 (Remarks to the Author)

Please see attached file.

Reviewer #2 (Remarks to the Author)

The submission describes the potential for bioaerosol generation by raindrops from soil. The study tests various soils and three different bacteria, then follows on with an estimate of how many bacteria may be aerosolised globally by this process.

The study reports only the impaction and growth of bacteria onto agar surfaces above soil as evidence for the bioaerosol generation, all other experimental work is based upon physical measurements and a part of this work has been previously published (see ref 20). In terms of measuring bacterial aerosolisation, the work ignores the major paradigm in microbiology that most microbes are unculturable and so measuring cultivable taxa like this is almost meaningless. As such there is really very little 'quantitative' data to support claims of microbial aerosol generation, and no consideration of other important soil microorganisms and propagules (for which you may expect differences based upon size if nothing else).

The environmental significance of this study is greatly over-stated (for the reasons stated above), and I am not convinced by the estimates for global aerosolisation of bacteria. Indeed the recent issues over revising estimates for the human microbiome highlight just how misleading these 'back of the envelope' calculations can be.

The brief consideration of how soil type may affect bacterial aerosolisation is not explored further, but would be interesting as an area to develop further in terms of explaining microbial dispersal (although this seems to be a major focus of the previous published study).

Overall I am also rather concerned that the manuscript implies transfer of pathogens from soil to air is the main ecological application for this work. It suggests a lack of awareness of the wider context for this work in terms of the major issues in microbial ecology of soils and the aerosphere (and the reference list reflects this).

Reviewer #3 (Remarks to the Author)

The article entitled "Bioaerosol generation by raindrops on soil" by Joung, et al. presents experimental evidence for a new mechanism by which they propose that large numbers of soil bacteria may be globally transmitted into the air. The authors present two supplemental high-speed videos that show ejection of particles from the surface of rain droplets impinging on a solid surface. They continue to discuss the viability of these bacterial particles and various other parameters of bacteria emission, including the rate of emission as a function of temperature and other factors. The videos are very interesting and the idea of bacterial transmission via this mechanism is tantalizing. I have not seen evidence of a mechanism quite like this, and indeed, the global impact of this process could be very important.

That said, my opinion is that the manuscript is not ready for publication in Nature Communications. The story is not sufficiently clear and the logic meanders a bit. Most importantly, there is very little discussion about the mechanism by which these bacteria are emitted or the reasons behind the observed relationships. My feeling is that these pieces of evidence could be cleaned up and presented more smoothly for a more technical journal whose readership is not the broad audience of NC. For this journal, however, a set of observations without explanation is likely

less interesting.

There are many plots that show relationships between number of emitted particles or viability versus other parameters, but there is almost no explanation given to discuss why these effects are present. For example, why is surface temperature so important to bacterial emission? Is it a physical effect, some adsorption constant, or is it a bacterial physiological effect?

I also suggest making a simple, clear story through which to present the various pieces of evidence. Individual figure panels are reasonably interesting, but the information presented once the panels are stitched together is less obvious. I got the feeling that the authors wanted to present a great deal more information than could be summarized in words quickly, and so each figure become a large, multi-panel presentation of information that was not well discussed.

The conclusions try to tie the story together into one with dramatic, global importance. This attempt skips from microscale observations of bacterial emission and makes sweeping statements of hope that are unfounded. For example, it is a huge step to say that key human infectious diseases they list might have some influence from the mechanism they report. For one thing, each of the diseases they list are viruses. The authors don't present any evidence for virus transmission by this mechanism, and it is not clear that the physical mechanism would even operate in this regime. So, while the idea is interesting, it is overly hypothetical.

In general, the experiments are interesting and likely worth publication somewhere, but I would recommend significant revisions before it be considered at Nature Communications.

Reviewer #4 (Remarks to the Author)

In 2014, Joung and Buie published in Nature Communications a paper on the mechanism of 'Aerosol generation by raindrop impact on soil'. In the submitted paper, they focus on bioaerosol and aim to demonstrate that bacteria can be released into the air from soil by raindrops and remain viable after aerosolisation. They combined different analytical techniques to evaluate the factors affecting this mechanism of transfer into the air and concluded about the impact of this mechanism in terms of dispersal of soil-borne bacteria on the planet.

The mechanisms enabling the aerosolisation of soil-borne microorganisms are not as well described as the bubble bursting at the air/water surface of oceans (Mayol et al., 2014, *Frontiers in Microbiology*, Volume 5 | Article 557; Walls et al., 2014; *Integrative and Comparative Biology*, doi:10.1093/icb/icu100...) and these research results on the effect of raindrops on bioaerosol generation from soil are novel and can have a significant impact for research people working on outdoor sources of bioaerosols. The multidisciplinary approach (physics of aerosol, microbiology) is also a source of novelty.

In particular, their key results are the following:

- The three bacteria tested could still be cultured after aerosolization, showing survival and potential transfer and dispersal from soil to air;
- Effect of the total volume of aerosols and of the number of bubbles on the number of dispersed particles; opposite trends depending on the origin of the bacteria (in the droplet or on soil) were observed;
- Impact of surface, impact velocity and surface temperature on particle transfer;
- Based on the results of the study, and estimation of the number of bacteria transferred from soil to air by raindrops was given: 10²² to 10²⁴ soil bacteria/year.

Concerning this last point, did the authors take into account the fact that rain will induce a 'washing out' of the aerosol due to collision phenomena?

The paper offers a detailed description of the mechanism of emission of bioaerosols from a surface due to raindrop and of the main factors affecting this process. The conclusion linking these results to the appearance of diseases near the equator is not appropriate for me.

The writing is not always clear, and I needed to read the previous paper to understand some parameters, material and methods chosen here. For example the sentence justifying the use of

TLC plate ('can be considered as ideal soil-like surface...') or the choice of rhodamine to demonstrate the presence of material in the droplet ('aerosol=suspension that contains chemical or solid particles') could be given again in this new paper. The paper from 2014 could be quoted in the Methods section, to explain how some parameters are measured (for example the impact speed, or the number of bubble and their diameter measured from high speed images).

Despite the real interest of the study, some points need to get clarified, especially concerning the set-up of experiments using bacterial suspension and microbiological analyses.

My main concern is on the data given on viability of bacteria.

First, viability is here examined by culturing microorganisms on plates. However, it is well known that some bacteria can lose culturability when submitted to environmental stress ('Viable But Non Culturable state'), and the fact that only a small percentage of airborne bacteria are culturable was previously published (review by Peccia and Hernandez in 2006; Atmospheric Environment 40, 3941-3961 giving culturability of airborne bacteria and fungi). The authors must discuss these aspects in the Discussion section.

To test viability, the sampling plates with collected bioaerosols were kept in the laboratory for a specific amount of time (first {section sign} of page 4). If I understand well, this is then the viability on sampling plate rather than the viability as airborne cells which was analysed. This is clearly not the same. The attachment to a surface probably affect the physiological state, and it is not the survival 'in air' which is here observed. For me the conclusion that 'the viability depend upon the residence time in the air' is not correct. If I misunderstood, please at least clarify the experimental set-up.

The surface (soil and TLC plates) pre-permeated with bacteria were dried for one day. Did the author check the effect of this treatment on the culturability/viability?

Were the soil sterilized (especially the one collected at Boston area?).

Also, were cells washed before being put in contact with soil or TLC plates. If it is not the case, could the volume of nutrient media (the volume is not specified) affect the surface properties and the aerosolisation?

Specify why this three bacterial species were chosen. If possible, specify the strain, since the type of strain can affect aerosolisation properties (see Gauthier-Levesque et al., 2016 in BMC Res Notes)

Method:

'cultured to the desired OD600'; authors should clarify the cell concentration and if cells were in exponential or stationary phase

Why putting the agar plates in a shaking incubator?

Statistics :

No statistics in the methods section.

In the first paragraph of the page 4, it is written 'The bacteria transferred to the aerosols exhibit different viability depending upon their residence time in the air, but there is no significant difference in the soil types and in the bacterial species' but there is no statistical analyses.

Concerning the effect of the surface temperature, it is written that 'At the surface temperature of 30 {degree sign}C, the least particles were dispersed for Case 1; however, the most particles were dispersed at the same temperature for Case 2.', but looking at the standard deviations on fig3c, this is not so obvious and results of statistical tests could be added in the text.

Writing :

'dispersion effectiveness' : this term was defined by the authors. In studies on bioaerosols, terms such as 'aerosolisation efficiency' were used (see Fahlgreen et al., Environmental Microbiology Reports (2015) 7(3), 460-470 for example), and the term 'aerosolisation' seems more adapted here than 'dispersion' as dispersion refers to processes that will affect the bioaerosol after its emission (dispersion by wind...).

In the discussion, the results are not put in perspective by quoting previous studies or giving the limitations/potential bias of the study. This needs to be done (see for example my remark concerning viability and culturability of bacteria).

Name of bacteria must be in italic.

Replace *Pseudomonas syringe* by *Pseudomonas syringae* in all the text

Use of present instead of past tense sometimes not adapted.

Fig4. Put the diamond symbol in the legend in green
Still several misspellings : 'sing drop'; 'florescent'; 'sampling pate'...

Response to Referees

Reviewer 1 (accept after revision)

Referee Report on: **Bioaerosol Generation by Raindrops on Soil**

By Joung, Ge and Buie

This manuscript describes experiments on the formation of aerosol containing bacteria by drops

impacting onto soils. The work shows convincingly that bacteria in the soil can be entrained into

splashed secondary droplets and carried aloft to become plausible spreading of disease. **The work needs some improvements, but I am supportive of acceptance after the following has been addressed.**

Comment A1: The authors mention “submicron” aerosols with reference to Figure 1(g). Compared to the scale bar of 50 μm , there are no dots less than 5 μm visible. The smallest one is 6 μm . Also in the abstract the authors talk about micrometer scale bubbles, when the visible bubbles in Figure 3(a) are probably dozens of microns. Please be more precise.

Response A1: Thank you for this comment and we apologize for the confusion. When we consider the images shown in Figure 1g and Figure 3a, the reviewer’s comment is correct. However, Figure 1(g) is a sample image of aerosols collected on the sampling plate. When we used a higher magnification microscope-lens, we observe submicron scale droplets on the plates (we didn’t show the images of the submicron aerosols in the manuscript). This size data can be seen in Figure 5d. Figure 5d shows the minimum diameter at 1 micrometer but this minimum value is the mean value including droplets in the submicron range. Considering the size of the aerosols, we can estimate the size of the bubbles that generated the aerosols. The bubble diameter is approximately ten times greater than the diameter of droplets generated as the bubble bursts [1, 2]; therefore, the minimum bubble size is a few microns. To clear up the confusion, we have added more details to captions of Figure 1g and Figure 3a in the manuscript.

Page 4 Column 1 Paragraph 1 Line 1: In Fig. 1g, the aerosol sizes range from a few microns to hundreds of microns; but with higher magnification we observe submicron droplets on the plates.

Page 6 Column 1 Paragraph 1 Line 9: On the clay and sandy clay soils, we observed more than one hundred aerosols smaller than 10 μm from a single drop impingement. This result indicates that micro bubbles are formed inside the raindrops hitting on the soils and the particles are dispersed when the bubbles burst at the surface of raindrops.

Abstract Line 7: Bubbles, tens of micrometers in size

Comment A2: Figure 2(c) shows the “average number of aerosols containing bacteria”. Majority of the values are less than 1. One would conclude that most drops don’t produce any bacteria emitted? Is this correct? Or should this be fraction of all splashed droplets which contain bacteria?

Response A2: Figure 2(c) shows the average number of colonies containing living bacteria from a single raindrop when the aerosol droplets, collected on the sampling plates, were transferred to the agar plates immediately after aerosolization. To clarify this point we have corrected the caption of Figure 2(c).

Figure 2: (b) Average number of colony forming units from a single raindrop when the aerosol droplets, collected on the sampling plates, were transferred to the agar plates immediately after aerosolization.

Comment A3: The caption of Figure 3 is somewhat confusing, as Case 1 and Case 2 are done with microparticles, but the caption talks about cells/nL. Why talk about cells if you are using micro-particles. Am I missing something here, or is the wording confused?

Response A3: Sorry for the confusion, we have changed the units from cells/nL to particles/nL.

Comment A4: More information is needed in some places.

1. What are the impact conditions in Figure 2b?
2. Similarly, what were the impact conditions for Figure 3(d)? Please add to caption.
3. I feel there is some ambiguity about the word aerosol. For example in Figure 1h, these are droplets with cells? The “red circles” are aerosols and “yellow dots” are bacteria. In this figure there are orange dots also. Perhaps say, large red circles are the aerosol droplets?
4. How is the evaporation of the droplets dealt with in the measurements of their sizes?

Response A4: We have now added the impact conditions to the captions of Figure 2b and Figure 3d (Figure 4 in the new version of manuscript). In Figure 1h, the red circle indicates an aerosol and the yellow-orange small dots indicate bacteria. To prevent this confusion, we have added arrows and captions indicating the aerosol and bacteria in Figures 1g-h. In this work, we didn't consider evaporation when calculating aerosol size because the time scale from the aerosol generation to the aerosol impact and measurement on the sampling plate is much smaller than the time scale of evaporation. The mean velocity of aerosol ejections is 10 m/s so the impact time scale is 1 microsecond because the gap distance from the surface to the sampling plate was 10 mm. With the consideration of the flying velocity and the mean aerosol size [3], under the common environment condition of 20 °C and 25 %RH, the change of diameter for an aerosol with initial diameter of 100 μm is less than 0.04 % so that the effect of evaporation on measured size is not significant. The evaporation is also occurred after the droplet sits on the sampling plate. It is known that the radius of droplets does not change during the evaporation if the contact line is pinned on the surface [4]. On hydrophilic surfaces, the contact line of water droplet is pinned on the surfaces. We used the acrylic sampling plates, which have the contact angle of $70.4 \pm 0.3^\circ$, and the constant droplet radius does not significantly change on the acrylic plates during the evaporation as reported in the other reference [5]. We have added text discussing the effect of evaporation on the droplet size in the method section on Page 10 Paragraph 2.

Figure 2: The impact velocity was 1.4 m/s with the raindrop diameter of 2.8 mm at the surface temperature of 20 °C for all the cases.

Figure 4: The impact velocity was 1.4 m/s with the raindrop diameter of 2.8 mm for all the surface temperatures.

Page 10 Column 1 Paragraph 2 Line 16: In this work, we did not consider evaporation when calculating aerosol size because the time scale from the aerosol generation to the aerosol impact and measurement on the sampling plate is much smaller than the time scale of evaporation. The mean velocity of aerosol ejections is 10 m/s and the gap distance from the surface to the sampling plate was 10 mm, so the impact time scale is 1 microsecond. Considering the flying velocity and the mean aerosol size [3] under the common environment condition of 20 °C and 25 %RH, the change of diameter for an aerosol with initial diameter of 100 μm is less than 0.04%, so the effect of evaporation on measured size is not significant.

Comment A5: On page 3: “The soils prepermeated with a red-fluorescence dye”. Please state that the soil is dry, this was not very clear?

Response A5: After permeation by red fluorescent dye, the soil was fully dried in a common laboratory environment. We have added description of the drying process to the manuscript on Page 3 Paragraph 2.

Page 3 Column 2 Paragraph 1 Line 2: The soils were fully dried under common laboratory conditions after pre-treatment with red-fluorescence dye.

Comment 6: X-axis of Figure 4(g): Impact velocity does not have units of m/s² !

Response 6: Thanks for catching this error, we have changed the units to m/s.

Comment 7: One limitation to the practicality of these experimental results is the fact that rain usually does not consist of only a few drops impacting a dry soil. After a few drops the soil is covered with a liquid film and the splashing changes drastically. This may not stop their aerosolization by rain-drop impacts, as the bacteria may reach the free surface and be splashed by later impacts. Thoroddsen *et al.*, *Phys. Rev. Lett.*, **106**, 034501 (2011) have shown that the ejecta sheet brings liquid from the bottom film and when it breaks up into fine spray it could act just like the proposed bubble mechanism. This should be mentioned.

Response 8: We appreciate the reviewer's comment on another potential mechanism of bacteria transfer by raindrop splashing. After multiple raindrops hit the same place on a soil surface, liquid films are formed on the surface and successive raindrops impact the film. Even though these raindrop impacts on the thin water film on soil would have a different mechanism from the impact on dry soil, as the reviewer mentioned, the material on the soil can be dispersed. We have cited the reference mentioned by the reviewer [6] and added a description of this potential mechanism of bacteria transfer from soil to air by splashing raindrops on wetted soil.

Page 4 Column 1 Paragraph 3 Line 5: The proposed mechanism is most relevant when the soil is not fully wet. After a couple of raindrops impact the same location, the soil is fully wet and a thin water film forms on the surface. In this case, we speculate that another possible mechanism of bacteria transfer by splashing is relevant. Thoroddsen *et al.* have shown that the ejected sheet brings liquid from the bottom of the film, and when it breaks up into fine spray it could act just like the proposed bubble mechanism [6]. If we consider this mechanism, it is possible that even more bacteria can be dispersed by raindrop impact. However, in the present work, we deal with the dispersion of bacteria by aerosolization [6].

Comment 8: The authors should not oversell the importance of the work to predict the spreading of disease. For example, in the last sentence of the Conclusions: MERS is primarily occurring in Saudi Arabia, where "high precipitation" does not play a role.

Response 8: We appreciate the reviewer's point here and have removed this expression and similar expressions from both the introduction and the conclusion. We have now added more broad applications of this work to the manuscript on Page 9 Paragraph 1.

Page 9 Column 1 Paragraph 2 Line 16: These findings reveal a novel mechanism of bacteria transfer from soil to the environment. This mechanism can be relevant for the investigation of climate change, pathogenic disease transmission, and geographic migration of bacteria.

Comment 9: Page numbers are missing in reference 20, which is the authors earlier paper related to this topic.

Response 9: We have added the page number to reference 20.

Comment 10: Figure 1(a-d): why not start the inset time t_{zero} at the first contact of the drop with the surface? That will give better feeling for the speed of the spreading in panel **b**.

Response 10: We have now changed the values of time in Figure 1(a-d) to show the time after drop impact.

Reviewer #2 (Remarks to the Author): (Minor revision)

The submission describes the potential for bioaerosol generation by raindrops from soil. The study tests various soils and three different bacteria, then follows on with an estimate of how many bacteria may be aerosolised globally by this process.

Comment B1: The study reports only the impaction and growth of bacteria onto agar surfaces above soil as evidence for the bioaerosol generation, all other experimental work is based upon physical measurements and a part of this work has been previously published (see ref 20).

Response B1: In this manuscript we verified bioaerosol generation with various quantitative data obtained with experiments that were not employed in our previous paper. In the present work, we mainly used microscopy of bioaerosols and viability testing of bioaerosols to characterize the mechanism that disperses bacteria into the air. The first method allows us to quantitatively characterize the amount of bacteria transferred by raindrop impact and the second method allows us to evaluate the effect of the aerosolization process on the viability of bacteria. In our previous work, high speed imaging was used to characterize the number of aerosols (not including bacteria) dispersed from raindrops on soil and the number of bubbles formed inside the impinging raindrop. In the present work, the number of bubbles formed inside a raindrop is the only physical parameter obtained by high-speed imaging and thus is similar to the previous study. Therefore, this work provides the new experimental methods to quantitatively characterize the properties of bioaerosols generated by raindrop impact on soil.

Comment B2: In terms of measuring bacterial aerosolisation, the work ignores the major paradigm in microbiology that most microbes are unculturable and so measuring cultivable taxa like this is almost meaningless. As such there is really very little 'quantitative' data to support claims of microbial aerosol generation, and no consideration of other important soil microorganisms and propagules (for which you may expect differences based upon size if nothing else).

Response B2: Thanks for the comment and we will try to clarify our position here. The objective of our viability test differs from general viability testing with soil bacteria. We intentionally used *cultured* soil bacteria to show that bacteria can survive after aerosolization. As the reviewer mentioned, if we use our viability test with the real soil containing various soil bacteria, we will observe fewer colonies because the majority of soil bacteria have not been cultured. Thus, we used direct visualization of the bioaerosols using microscopy. Using this visualization method, we can measure the number of bacteria transferred by raindrop impact regardless of the culturability of soil bacteria. Further, we disagree with the reviewer's comments on the lack of quantitative data because we quantify the number of bacteria transferred by aerosols with respect to the surface bacteria density, surface temperature, impact conditions, and soil type. Table 1 and Figure 2 and Figure 5 show the quantitative data of bacteria transferred by raindrop impact. It should be noted that the inability to culture an organism in the lab does not immediately suggest the organism isn't viable, as the majority of bacteria on the planet have not been cultivated in the lab. To further clarify the manuscript, we have added discussion illuminating the limitations of the viability test we used to the manuscript on Page 4, Paragraph 1.

Page 4 Column 1 Paragraph 2 Line 23: The viability test used in this work may not be applicable for other soil bacteria that may lose their culturability when subjected to environmental stresses or an inadequate cultivation environment. Indeed, it is well known that a small percentage of airborne bacteria can be cultured [7, 8]. In fact most estimates suggest that the majority of bacteria on the planet have evaded laboratory cultivation [9-12].

In this work, we intentionally seeded cultured bacteria into the soil before aerosolization to check if the soil bacteria can survive the aerosolization process. We did not use bacteria directly recovered from environmental samples transferred by aerosols. If the aerosolization results in some bacteria remaining viable but unculturable, the number of bacteria transferred through aerosolization would be underestimated. However, this underestimation does not undermine the main result of our viability test. To minimize the limitations of viability testing, we used direct visualization of aerosols containing bacteria to count the number of bacteria transferred by aerosols; therefore, the aerosolization efficiency reported is not distorted by bacteria culturability.

Comment B3: The environmental significance of this study is greatly over-stated (for the reasons stated above), and I am not convinced by the estimates for global aerosolization of bacteria. Indeed the recent issues over revising estimates for the human microbiome highlight just how misleading these 'back of the envelope' calculations can be.

Response B3: We appreciate the comment and as the reviewer mentioned, it is challenging to estimate the global transfer of soil bacteria by raindrops due to uncertainties in global climate, soil type, washout, and soil bacteria species. Thus, we mentioned this estimation in the discussion section (not in the result section) with the minimum and maximum order of magnitudes of the global bacteria transmission by raindrops. We discussed the potential global transfer amount to emphasize the potential implications of this work. However, we understand the reviewer's comment and we have now moved the estimation of global transfer of bacteria by raindrops to the supplementary information. In addition, we have added discussion of the limitations to our estimate and possible mechanisms that would reduce the actual transfer amount to the manuscript on Page 8 Paragraph 1.

Page 8 Column 2 Paragraph 1 Line 10: However, this estimate should be considered an upper bound because successive raindrops can wash the aerosols out of the air. As discussed in our previous work [13], however, aerosols can be convectively migrated by wind, and thus it is expected that a reasonable fraction of bioaerosols can be dispersed into the atmosphere and evade washout by successive raindrops. Others have shown that aerosol concentration in the atmosphere rapidly increases after rainfall, suggesting that washout does not remove all aerosols.[14-16]

Comment B4: The brief consideration of how soil type may affect bacterial aerosolization is not explored further, but would be interesting as an area to develop further in terms of explaining microbial dispersal (although this seems to be a major focus of the previous published study).

Response B4: As the reviewer mentioned, in our previous work, we focused on the influence of soil type on aerosolization. In this work, we sought to quantitatively characterize the amount of bacteria transferred by raindrop impact on three representative soil types: clay, sandy clay, and sand. We agree with the reviewer's comment and have added a description on the effect of soil type on bioaerosol generation.

Page 7 Column 2 Paragraph 3 Line 1: The number of aerosols containing bacteria varies with soil type. Sandy clay soils showed the most bioaerosols while sandy soils don't appear to produce bioaerosols. This result is consistent with our previous work showing that more aerosols can be generated on surfaces with wetting properties similar to sandy clay soils. Sandy clay soils have ideal properties for aerosol generation and show the highest aerosol generation relative to the other soils tested, as shown in Figure 6. Sandy soils have fast water absorption speeds such that aerosols are not generated by raindrop impact because bubbles do not form.

Comment B5: Overall I am also rather concerned that the manuscript implies transfer of pathogens from soil to air is the main ecological application for this work. It suggests a lack of awareness of the wider context for this work in terms of the major issues in microbial ecology of soils and the aerosphere (and the reference list reflects this).

Response B5: Thanks for the comment and we have modified the language in alignment with this point. Considering the reviewer's comment, we have updated the Abstract, Introduction, and Results to include the wider implication of this work, de-emphasizing pathogen transfer, and have added related references.

Abstract: Aerosolized microorganisms play an important role in climate change, disease transmission, water and ground contamination, and geographic migration of microbes.

Page 2 Column 1 Paragraph 1 Line 2: In the atmosphere, bioaerosols have critical impact on the global climate, promoting cloud formation and ice nucleation even though their fraction is relatively small to all the atmospheric aerosols [14, 17-20]. On the ground, bioaerosols can change micro-biogeography faster than many other transport mechanisms [21, 22]. Because bioaerosols can lead to dispersion of biological contaminants over long distances relative to terrestrial transport mechanisms [23, 24], they significantly affects the change of biodiversity and ecology as well as the propagation of biological pollutants [20, 25]. Furthermore, bioaerosols can be effective carriers of pathogenic organisms to plants, animals, and humans, resulting in the fast and wide spread of disease [26, 27].

Page 9 Column 1 Paragraph 2 Line 16: These findings reveal a novel mechanism of bacteria transfer from soil to the environment. This mechanism can be relevant for the investigation of climate change, pathogenic disease transmission, and geographic migration of bacteria.

Reviewer #3 (Remarks to the Author): (Major revision)

The article entitled "Bioaerosol generation by raindrops on soil" by Joung, et al. presents experimental evidence for a new mechanism by which they propose that large numbers of soil bacteria may be globally transmitted into the air. The authors present two supplemental high-speed videos that show ejection of particles from the surface of rain droplets impinging on a solid surface. They continue to discuss the viability of these bacterial particles and various other parameters of bacteria emission, including the rate of emission as a function of temperature and other factors. The videos are very interesting and the idea of bacterial transmission via this mechanism is tantalizing. I have not seen evidence of a mechanism quite like this, and indeed, the global impact of this process could be very important.

Comment C1: That said, my opinion is that the manuscript is not ready for publication in Nature Communications. The story is not sufficiently clear and the logic meanders a bit. Most importantly, there is very little discussion about the mechanism by which these bacteria are emitted or the reasons behind the observed relationships. My feeling is that these pieces of evidence could be cleaned up and presented more smoothly for a more technical journal whose readership is not the broad audience of NC. For this journal, however, a set of observations without explanation is likely less interesting. There are many plots that show relationships between number of emitted particles or viability versus other parameters, but there is almost no explanation given to discuss why these effects are present. For example, why is surface temperature so important to bacterial emission? Is it a physical effect, some adsorption constant, or is it a bacterial physiological effect?

Response C1: Thank you for your comment and enthusiasm about this manuscript. In this work, we focused on bubble formation and bursting as the main mechanism of bacteria transfer by raindrop impact. We show that bubble bursting inside a raindrop is the dominant mechanism of bacteria transfer through the bioaerosols. The number of bioaerosols can be expressed as a function of the number of bubbles formed inside the impinging raindrop. We show that the number of aerosols governs the amount of bacteria transferred by a single raindrop. The effect of the surface temperature and impact condition is explained relative to the number and size of bubbles in the raindrop. We employ the term "*aerosolization efficiency*" to characterize how many bacteria can be transferred by bubbles generated by a single raindrop impact. As the reviewer mentioned, the transfer effectiveness is potentially affected by the physicochemical properties of bacteria and the interaction of bacteria and surfaces. However, the investigation of the effect of physicochemical properties is beyond the present work and is worthy of an entirely independent study. We believe that this work provides clear evidence of bioaerosol generation by raindrop impact and future work will investigate the reasons different combinations of bacteria and soil result in varying amounts of dispersion.

Comment C2: I also suggest making a simple, clear story through which to present the various pieces of evidence. Individual figure panels are reasonably interesting, but the information presented once the panels are stitched together is less obvious. I got the feeling that the authors wanted to present a great deal more information than could be summarized in words quickly, and so each figure become a large, multi-panel presentation of information that was not well discussed.

Response C2: In an effort to make the manuscript simple and clear, we have separated Figure 3 (in the old version) into Figures 3 and 4 (in the new version) and Figure 4 (in the old version) into Figures 5 and 6 (in the new version) with more additional description in the figure captions and the main text.

Comment C3: The conclusions try to tie the story together into one with dramatic, global importance. This attempt skips from microscale observations of bacterial emission and makes sweeping statements of hope that are unfounded. For example, it is a huge step to say that key human infectious diseases they list might have some influence from the mechanism they report. For one thing, each of the diseases they list are viruses. The authors don't present any evidence for virus transmission by this mechanism, and it is not clear that the physical mechanism would even operate in this regime. So, while the idea is interesting, it is overly hypothetical.

Response C3: We agree with the reviewer's comment and have updated the introduction and conclusion to exclude overstatements of the potential impact of this work. Instead we have added more broad applications of this work in the abstract, introduction, and results sections.

Abstract Line 1: Aerosolized microorganisms play an important role in climate change, disease transmission, water and ground contamination, and geographic migration of microbes.

Page 2 Column 1 Paragraph 1 Line 2: In the atmosphere, bioaerosols have critical impact on the global climate, promoting cloud formation and ice nucleation even though their fraction is relatively small to all the atmospheric aerosols [14, 17-20]. On the ground, bioaerosols can change micro-biogeography faster than many other transport mechanisms [21, 22]. Because bioaerosols can lead to dispersion of biological contaminants over long distances relative to terrestrial transport mechanisms [23, 24], they significantly affects the change of biodiversity and ecology as well as the propagation of biological pollutants [20, 25]. Furthermore, bioaerosols can be effective carriers of pathogenic organisms to plants, animals, and humans, resulting in the fast and wide spread of disease [26, 27].

Page 9 Column 1 Paragraph 2 Line 16: These findings reveal a novel mechanism of bacteria transfer from soil to the environment. This mechanism can be relevant for the investigation of climate change, pathogenic disease transmission, and geographic migration of bacteria.

In general, the experiments are interesting and likely worth publication somewhere, but I would recommend significant revisions before it be considered at Nature Communications.

Reviewer #4 (Remarks to the Author): (Minor revision)

In 2014, Joung and Buie published in Nature Communications a paper on the mechanism of 'Aerosol generation by raindrop impact on soil'. In the submitted paper, they focus on bioaerosol and aim to demonstrate that bacteria can be released into the air from soil by raindrops and remain viable after aerosolisation. They combined different analytical techniques to evaluate the factors affecting this mechanism of transfer into the air and concluded about the impact of this mechanism in terms of dispersal of soil-borne bacteria on the planet.

The mechanisms enabling the aerosolisation of soil-borne microorganisms are not as well described as the bubble bursting at the air/water surface of oceans (Mayol et al., 2014, *Frontiers in Microbiology*, Volume 5 | Article 557; Walls et al., 2014; *Integrative and Comparative Biology*, doi:10.1093/icb/icu100...) and these research results on the effect of raindrops on bioaerosol generation from soil are novel and can have a significant impact for research people working on outdoor sources of bioaerosols. The multidisciplinary approach (physics of aerosol, microbiology) is also a source of novelty.

In particular, their key results are the following: - The three bacteria tested could still be cultured after aerosolization, showing survival and potential transfer and dispersal from soil to air; - Effect of the total volume of aerosols and of the number of bubbles on the number of dispersed particles; opposite trends depending on the origin of the bacteria (in the droplet or on soil) were observed; - Impact of surface, impact velocity and surface temperature on particle transfer; - Based on the results of the study, and estimation of the number of bacteria transferred from soil to air by raindrops was given: 10^{22} to 10^{24} soil bacteria/year.

Comment D1: Concerning this last point, did the authors take into account the fact that rain will induce a 'washing out' of the aerosol due to collision phenomena?

Response D1: We didn't consider the washing out of aerosols by successive raindrops in the estimation of global transmission of bacteria by raindrop impact. However, we dealt with the same issue in our previous work [13] (Aerosol generation by raindrop impact on soil) with possible mechanisms which can avoid the removal process to the manuscript. To address the reviewer's comment, we have altered the language outlining our estimate to discuss the washing out process on Page 7 Paragraph 2.

Page 8 Column 2 Paragraph 1 Line 7: However, this estimate should be considered an upper bound because successive raindrops can wash the aerosols out of the air. As discussed in our previous work [13], however, aerosols can be convectively migrated by wind, and thus it is expected that a reasonable fraction of bioaerosols can be dispersed into the atmosphere and evade washout by successive raindrops. Others have shown that aerosol concentration in the atmosphere rapidly increases after rainfall, suggesting that washout does not remove all aerosols.[14-16]

In our previous paper:

Page 7 and Paragraph 1: In this work we demonstrate that aerosols can be generated from raindrops on soil. However, there are a few limitations of this work that have yet to be fully explored. First, raindrop-aerosol collisions are a major mechanism of aerosol removal³. Even though aerosols are generated from droplets, successive raindrops can eliminate the aerosols from the air; therefore, large scale environmental effects might not be expected. Second, the gravitational settling velocity can be relatively fast when the droplet size is around a few tens of microns. Even when considering evaporation, the settling speed is not negligible.

Therefore, further investigation of aerosol migration is necessary to understand the potential for large scale environmental effects of aerosols generated on soil. One mechanism that could enhance the relevance of this phenomenon is wind driven advection. Considering wind on the surface of porous media, tiny droplets or solid particles ($< 50 \mu\text{m}$) can be transferred very long distances from the origin, up to a few thousand kilometers^{61,62}. Furthermore, aerosols can have higher velocities than raindrops due to their small size, leading to lower drag forces^{24,30}. Therefore, aerosols have the potential to migrate long distances without serious washing out or gravitational settling. In this manuscript, we focused on a novel mechanism of aerosol generation from droplets hitting porous media rather than exploring the effects of the aerosols. This discovery will lead to additional work investigating the fate of these aerosols in the environment.

Comment D2: The paper offers a detailed description of the mechanism of emission of bioaerosols from a surface due to raindrop and of the main factors affecting this process. The conclusion linking these results to the appearance of diseases near the equator is not appropriate for me.

Response D2: We agree with the reviewer's comment and we have removed some of these comments from the results/conclusions. Follow up work is necessary to connect the result of this paper to the appearance of diseases to specific regions.

Comment D3: The writing is not always clear, and I needed to read the previous paper to understand some parameters, material and methods chosen here. For example the sentence justifying the use of TLC plate ('can be considered as ideal soil-like surface...') or the choice of rhodamine to demonstrate the presence of material in the droplet ('aerosol=suspension that contains chemical or solid particles') could be given again in this new paper. The paper from 2014 could be quoted in the Methods section, to explain how some parameters are measured (for example the impact speed, or the number of bubble and their diameter measured from high speed images). Despite the real interest of the study, some points need to get clarified, especially concerning the set-up of experiments using bacterial suspension and microbiological analyses.

Response D3: We appreciate the reviewer's kind and careful reading of the manuscript. We have added more detailed information about the materials and methods used in this work.

On Page 2 Column 2 Paragraph 2 Line 1: Aerosols represent small water droplets containing chemicals or solid particles; while bioaerosols contain microbes.

Caption of Figure 1 Line 2: The TLC plate served as an ideal soil-like surface.

Page 9 Column 2 Paragraph 3 Line 2: Before treatment of soils with microspheres or bacteria, the soils were sterilized and dried for a day. To prepare soils pre-permeated with microspheres or bacteria, the microsphere or bacterial suspensions were placed on the soil surface with a specific volume (0.05 mL/cm^2). To prepare TLC plates permeated with microspheres or bacteria, the TLC plates were soaked in microsphere suspensions for 30 seconds and then dried for 25 min in a common laboratory environment. The surface particle or cell density was measured with digital microscopy.

Page 10 Column 1 Paragraph 3 Line 13: The impact speed was estimated using the drop release height and calibrated with high speed movies. We used a high-speed camera (Photron, San Diego, California, USA) to record the process of drop impingement, bubble formation, and bubble bursting inside the droplets. The diameter and number of bubbles were measured by an imaging analysis program (ImageJ [28, 29]) applied to the high speed movies.

Comment D4: My main concern is on the data given on viability of bacteria. First, viability is here examined by culturing microorganisms on plates. However, it is well known that some

bacteria can lose culturability when submitted to environmental stress ('Viable But Non Culturable state'), and the fact that only a small percentage of airborne bacteria are culturable was previously published (review by Peccia and Hernandez in 2006; Atmospheric Environment 40, 3941-3961 giving culturability of airborne bacteria and fungi). The authors must discuss these aspects in the Discussion section.

Response D4: The reviewer's comment is correct and we have added text outlining the difficulty in culturing bacteria after they are exposed to the environmental stress along with the corresponding reference [7] the reviewer mentioned. In this work, we are intentionally seeding soil with culturable microbes to determine if soil bacteria could potentially survive the aerosolization process. To minimize contamination the soil is sterilized before we add our cultivable strains. In this first study we deliberately do not test microbes directly recovered from environmental samples to circumvent cultivability. The biggest limitation of our current method is if the aerosolization renders some bacteria viable yet uncultivable (though they originate from cultivable strains). In this case we would be underestimating the number of microbes transferred through aerosolization. Therefore, our viability data represents a minimum number of bacteria that can be transferred. Further, the results show that the aerosolization process does not generate serious stress since we can measure cultivable microbes. To clarify the text on this subject we have added the limitations of the viability test we used to the manuscript on Page 4 Paragraph 1.

Page 4 Column 1 Paragraph 2 Line 23: The viability test used in this work may not be applicable for other soil bacteria that may lose their culturability when subjected to environmental stresses or an inadequate cultivation environment. Indeed, it is well known that a small percentage of airborne bacteria can be cultured [7, 8]. In fact most estimates suggest that the majority of bacteria on the planet have evaded laboratory cultivation [9-12]. In this work, we intentionally seeded cultured bacteria into the soil before aerosolization to check if the soil bacteria can survive the aerosolization process. We did not use bacteria directly recovered from environmental samples transferred by aerosols. If the aerosolization results in some bacteria remaining viable but unculturable, the number of bacteria transferred through aerosolization would be underestimated. However, this underestimation does not undermine the main result of our viability test. To minimize the limitations of viability testing, we used direct visualization of aerosols containing bacteria to count the number of bacteria transferred by aerosols; therefore, the aerosolization efficiency reported is not distorted by bacteria culturability.

Comment D5: To test viability, the sampling plates with collected bioaerosols were kept in the laboratory for a specific amount of time (first section of page 4). If I understand well, this is then the viability on sampling plate rather than the viability as airborne cells which was analyzed. This is clearly not the same. The attachment to a surface probably affect the physiological state, and it is not the survival 'in air' which is here observed. For me the conclusion that 'the viability depend upon the residence time in the air' is not correct. If I misunderstood, please at least clarify the experimental set-up.

Response D5: The reviewer is correct in that we are testing viability on the sampling plates. We emphasized the parameters of the viability test in the caption of Figure 2 as follows: "(c) viability test with respect to the duration of drying the aerosols collected on the sampling plates. The times, displayed in the images, indicate the drying duration. We used the sampling plates to collect the bioaerosol and they were maintained on the plates during a specific time to check their viability." As the reviewer notes, this condition does not provide the same environment as dispersal in the air. As a result, we have changed the term "in air" to "on the surface exposed to the air". The main objective of this viability test is the verification

of bacteria survival during the aerosolization process, thus the use of sampling plates is still relevant.

Comment D6: The surface (soil and TLC plates) pre-permeated with bacteria were dried for one day. Did the author check the effect of this treatment on the culturability/viability?

Response D6: We apologize for the confusion but the reviewer may have misinterpreted a portion of the text. The soils were dried for one day before the bacteria were added. Then, the soils were pre-permeated with bacteria and kept in a common laboratory environment for 25 minutes for drying. We found that the wetting properties of the soils were fully recovered within 25 minutes after seeding the bacteria. We tested the viability of bacteria on the permeated soils with both soil bacteria such as *C. glutamicum* and non-soil bacteria (*E.coli*). Soil bacteria can survive during the drying time, while viable *E. coli* were not observed. We have now clarified the experimental conditions for the viability test in the Methods section on Page 10 Paragraph 4.

Page 10 Column 2 Paragraph 2 Line 7: We found that the *C. glutamicum* can survive on the soils and TLC-plates after drying. In order to test cell viability after aerosolization, the soils and TLC plates were permeated with bacterial suspensions and kept in a common laboratory environment for 25 minutes prior to aerosolization experiments.

Comment D7: Were the soil sterilized (especially the one collected at Boston area?).

Response D7: Yes, they were all sterilized.

Comment D8: Also, were cells washed before being put in contact with soil or TLC plates. If it is not the case, could the volume of nutrient media (the volume is not specified) affect the surface properties and the aerosolization?

Response D8: We diluted the cells directly starting with nutrient media. After the surfaces were permeated with the cell suspension we checked aerosolization process after different amounts of drying time. We found that the original wetting properties of the surfaces were fully recovered within 10 minutes after permeation. To be careful, we dried the surfaces for 25 minutes to prevent changes in wetting properties of the soils and TLC plates during aerosolization experiments. We have added the description above in the manuscript on Page 10 Paragraph 4.

Page 10 Column 2 Paragraph 2 Line 12: We diluted the cells directly starting with nutrient media. After the surfaces were permeated with the cell suspension we checked aerosolization process after different amounts of drying time. We found that the original wetting properties of the surfaces were fully recovered within 10 minutes after permeation. To be careful, we dried the surfaces for 25 minutes to prevent changes in wetting properties of the soils and TLC plates during aerosolization experiments.

Comment D9: Specify why this three bacterial species were chosen. If possible, specify the strain, since the type of strain can affect aerosolisation properties (see Gauthier-Levesque et al., 2016 in BMC Res Notes)

Response D9: The exact bacterial strains are specified in the Methods section, but we have now added this information to the main text. We used the three species of soil bacteria because they are all abundant in the soil, and maintain viability/cultivability on the surfaces of soils and TLC-plates used in this work. We tested the viability of bacteria on the permeated soils with both soil bacteria such as *C. glutamicum* and non-soil bacteria such as *E.coli*. We found that the soil bacteria can survive after the surfaces were dried while *E. coli* cannot. We have added the reason we used the three species represented in this work to the

manuscript on Page 3 Paragraph 2 while also citing the reference [30] the reviewer mentioned.

Page 3 Column 2 Paragraph 1 Line 12: Depending on the species and strains of bacteria, it is known that bacteria show different aerosolization properties [30]. In this work, we used three representative bacteria that are each abundant in soil bacteria and show long-term viability on soils and TLC-plates. These strains allow us to evaluate the effect of aerosolization on cell viability without considerable numbers of dead cells on the surfaces.

Comment D10: Method: 'cultured to the desired OD600'; authors should clarify the cell concentration and if cells were in exponential or stationary phase.

Response D10: Because the OD600 for the three types of cells tested is slightly different, we use the word “desired” here. The exact OD600 is specified in the main text but all cells are in exponential phase. To calibrate this desired cell concentration, we used microscopy images of cells on the surfaces to directly measure the cell density on the surface.

Comment D11: Why putting the agar plates in a shaking incubator?

Response D11: As the reviewer suggests, agar plates don't need to be put in a shaking incubator, this is done due to equipment available in our lab. We use shaking incubators for both culturing cells and incubating agar plates.

Comment D7: Statistics: No statistics in the methods section. In the first paragraph of the page 4, it is written 'The bacteria transferred to the aerosols exhibit different viability depending upon their residence time in the air, but there is no significant difference in the soil types and in the bacterial species' but there is no statistical analyses.

Response D7: Thanks for this comment. We have now added the result of the viability test statistical analysis to the manuscript on Page 4 Paragraph 1. In addition, the methods used for the statistical analysis have been added to the methods section of the manuscript.

Methods: The viability data were obtained with three different species of bacteria and five types of soil in 15 minute intervals for 60 min. A regression analysis was performed to identify any correlation between drying time and viability with a 95% confidence level (when $p < 0.05$). The Student's T-test was used to identify significant differences between soil types and bacterial species with a 95% confidence level (when $p < 0.05$).

Page 4 Column 1 Paragraph 2 Line 14: The bacteria transferred to the aerosols exhibit different viability depending upon their residence time in the air ($p < 0.05$), but there is no significant difference in the soil types and in the bacterial species ($p > 0.05$).

Comment D8: Concerning the effect of the surface temperature, it is written that 'At the surface temperature of 30 °C, the least particles were dispersed for Case 1; however, the most particles were dispersed at the same temperature for Case 2.', but looking at the standard deviations on fig3c, this is not so obvious and results of statistical tests could be added in the text.

Response D8: Because we used the log scale in y-axis of the graph in Figure 3c, the difference looks insignificant. To clarify the confusion, we have now changed the graph from a semi logarithmic scale to a linear scale. In addition, we have added the statistical test used to verify the different trends of particle dispersion for both cases. We performed a T-test to identify significant differences between Case 1 and Case 2 for the different concentrations with a 95% confidence level. We have now added the statistical result to Page 4 Paragraph 2.

Page 5 Column 2 Paragraph 1 Line 3: At the surface temperature of 30 °C, the least particles were dispersed for Case 1 ($p < 0.05$); however, the most particles were dispersed at the same temperature for Case 2 ($p < 0.05$).

Comment D9: Writing: 'dispersion effectiveness' : this term was defined by the authors. In studies on bioaerosols, terms such as 'aerosolisation efficiency' were used (see Fahlgreen et al., Environmental Microbiology Reports (2015) 7(3), 460-470 for example), and the term 'aerosolisation' seems more adapted here than 'dispersion' as dispersion refers to processes that will affect the bioaerosol after its emission (dispersion by wind...).

Response D9: Thanks for this comment, hopefully this will make our language more consistent with the current literature. In the updated manuscript we now use “aerosolization efficiency” instead of “dispersion effectiveness”, citing reference [31] the reviewer mentioned throughout the manuscript.

Page 5 Column 2 Paragraph 4 Line 1: We employ *aerosolization efficiency* [31] (ϵ) to characterize how many particles or bacteria can be transferred by a single raindrop from soil to air.

Comment D10: In the discussion, the results are not put in perspective by quoting previous studies or giving the limitations/potential bias of the study. This needs to be done (see for example my remark concerning viability and culturability of bacteria).

Response D10: We have now added description of the limitations/potential bias of the study, mainly focusing on the viability and culturability the reviewer mentioned as following:

Page 4 Column 1 Paragraph 2 Line 23: The viability test used in this work may not be applicable for other soil bacteria that may lose their culturability when subjected to environmental stresses or an inadequate cultivation environment. Indeed, it is well known that a small percentage of airborne bacteria can be cultured [7, 8]. In fact most estimates suggest that the majority of bacteria on the planet have evaded laboratory cultivation [9-12]. In this work, we intentionally seeded cultured bacteria into the soil before aerosolization to check if the soil bacteria can survive the aerosolization process. We did not use bacteria directly recovered from environmental samples transferred by aerosols. If the aerosolization results in some bacteria remaining viable but unculturable, the number of bacteria transferred through aerosolization would be underestimated. However, this underestimation does not undermine the main result of our viability test. To minimize the limitations of viability testing, we used direct visualization of aerosols containing bacteria to count the number of bacteria transferred by aerosols; therefore, the aerosolization efficiency reported is not distorted by bacteria culturability.

Comment D11: Name of bacteria must be in italic. Replace *Pseudomonas syringe* by *Pseudomonas syringae* in all the text. Use of present instead of past tense sometimes not adapted.

Fig4. Put the diamond symbol in the legend in green. Still several misspellings: 'sing drop'; 'fiorescent'; 'sampling pate'...

Response D11: Thank you for catching this, we have corrected bacterial names in italics and changed *Pseudomonas syringe* into *Pseudomonas syringae*. We have used present tense instead of past tense on Page 3 Paragraphs 2, Page 4 Paragraph 1, Page 4 Paragraph 2, and Page 5 Paragraph 2. We have also corrected Figure 4 (in the current version, Figures 5d and e). Finally, we have corrected the typos the reviewer's mentioned. We sincerely appreciate the reviewer's kind proof reading.

References

1. Kientzler, C.F., et al., *Photographic investigation of the projection of droplets by bubbles bursting at a water surface*. *Tellus*, 1954. **6**(1): p. 1-7.
2. Wu, J.I.N., *Contributions of film and jet drops to marine aerosols produced at the sea surface*. *Tellus B*, 1989. **41B**(4): p. 469-473.
3. Geng, Y., et al., *Effects of surfactant treatment on mechanical and electrical properties of CNT/epoxy nanocomposites*. *Composites Part A: Applied Science and Manufacturing*, 2008. **39**(12): p. 1876-1883.
4. Yarin, A.L., *DROP IMPACT DYNAMICS: Splashing, Spreading, Receding, Bouncing...* *Annual Review of Fluid Mechanics*, 2006. **38**(1): p. 159-192.
5. Thoroddsen, S.T., et al., *The air bubble entrapped under a drop impacting on a solid surface*. *J.Fluid Mech.*, 2005. **545**: p. 203-212.
6. Thoroddsen, S.T., et al., *Droplet splashing by a slingshot mechanism*. *Physical Review Letters*, 2011. **106**(3): p. 034501.
7. Peccia, J. and M. Hernandez, *Incorporating polymerase chain reaction-based identification, population characterization, and quantification of microorganisms into aerosol science: A review*. *Atmospheric Environment*, 2006. **40**(21): p. 3941-3961.
8. Fierer, N., et al., *Short-term temporal variability in airborne bacterial and fungal populations*. *Applied and Environmental Microbiology*, 2008. **74**(1): p. 200-207.
9. Stevens, B. and O. Boucher, *Climate science: the aerosol effect*. *Nature*, 2012. **490**(7418): p. 40-41.
10. Carslaw, K.S., et al., *Large contribution of natural aerosols to uncertainty in indirect forcing*. *Nature*, 2013. **503**(7474): p. 67-71.
11. Myhre, G., et al., *Aerosols and their relation to global climate and climate sensitivity*. *Nat. Educ. Knowl.*, 2013. **4**(5): p. 7.
12. Paasonen, P., et al., *Warming-induced increase in aerosol number concentration likely to moderate climate change*. *Nat. Geosci.*, 2013. **6**(6): p. 438-442.
13. Joung, Y.S. and C.R. Buie, *Aerosol generation by raindrop impact on soil*. *Nature Communications*, 2015. **6**.
14. Rastogi, R., et al., *Comparative study of carbon nanotube dispersion using surfactants*. *Journal of Colloid and Interface Science*, 2008. **328**(2): p. 421-428.
15. Constantinidou, H., et al., *Atmospheric dispersal of ice nucleation-active bacteria: the role of rain*. *Phytopathology*, 1990. **80**(10): p. 934-937.
16. Christner, B.C., et al., *Ubiquity of biological ice nucleators in snowfall*. *Science*, 2008. **319**(5867): p. 1214.
17. Szyrmer, W. and I. Zawadzki, *Biogenic and anthropogenic sources of ice-forming nuclei: a review*. *Bulletin of the American Meteorological Society*, 1997. **78**(2): p. 209-228.
18. Bowers, R.M., et al., *Characterization of airborne microbial communities at a high-elevation site and their potential to act as atmospheric ice nuclei*. *Applied and Environmental Microbiology*, 2009. **75**(15): p. 5121-5130.
19. Lindemann, J., et al., *Plants as sources of airborne bacteria, including ice nucleation-active bacteria*. *Applied and Environmental Microbiology*, 1982. **44**(5): p. 1059-1063.
20. Villermaux, E. and B. Bossa, *Single-drop fragmentation determines size distribution of raindrops*. *Nat. Phys.*, 2009. **5**(9): p. 697-702.

21. Abu-Ashour, J., et al., *Transport of microorganisms through soil*. Water, Air, and Soil Pollution, 1994. **75**(1): p. 141-158.
22. Fierer, N., *Microbial Biogeography: Patterns in Microbial Diversity across Space and Time*, in *Accessing Uncultivated Microorganisms*, K. Zengler, Editor 2008, American Society of Microbiology: Washington DC. p. 95-115.
23. Teltsch, B. and E. Katzenelson, *Airborne enteric bacteria and viruses from spray irrigation with wastewater*. Applied and Environmental Microbiology, 1978. **35**(2): p. 290-296.
24. Nhu Nguyen, T.M., et al., *A community-wide outbreak of legionnaires disease linked to industrial cooling towers—how far can contaminated aerosols spread?* Journal of Infectious Diseases, 2006. **193**(1): p. 102-111.
25. Ahn, S., et al., *Effects of hydrophobicity on splash erosion of model soil particles by a single water drop impact*. Earth Surf. Proc. Land., 2013. **38**(11): p. 1225-1233.
26. Douwes, J., et al., *Bioaerosol health effects and exposure assessment: progress and prospects*. Annals of Occupational Hygiene, 2003. **47**(3): p. 187-200.
27. Cooke, W.B., *The air spora*. Ecology, 1964. **45**(1): p. 212-213.
28. Marston, J.O., et al., *Experimental study of liquid drop impact onto a powder surface*. Powder Technol., 2010. **203**(2): p. 223-236.
29. Nefzaoui, E. and O. Skurtys, *Impact of a liquid drop on a granular medium: Inertia, viscosity and surface tension effects on the drop deformation*. Exp. Therm. Fluid Sci., 2012. **41**(0): p. 43-50.
30. Gauthier-Levesque, L., et al., *Impact of serotype and sequence type on the preferential aerosolization of Streptococcus suis*. BMC Research Notes, 2016. **9**: p. 273.
31. Fahlgren, C., et al., *Seawater mesocosm experiments in the Arctic uncover differential transfer of marine bacteria to aerosols*. Environmental Microbiology Reports, 2015. **7**(3): p. 460-470.

Reviewers' Comments:

Reviewer #3 (Remarks to the Author)

The article entitled "Bioaerosol Generation by Raindrops on Soil" submitted by Joung, Ge, and Buie has been revised after taking comments from reviewers into account. After re-reading it, my feel is that the manuscript has made some improvements, especially by removing over-stated thoughts about potential application to global health. The manuscript is interesting and presents interesting experimental work. Overall, however, my feel is that the message is not crisply presented and would benefit from significant streamlining of the plots and associated message. Several figures have been chopped into multiple figures, but the total number of plots appears not to have changed. Along with this, there are so many different stories and messages presented by each figure that it is difficult to follow along in detail. There is simply not enough room in the short article to describe in text about all the messages that authors attempt to show in the figures. My feeling is that the manuscript is still too full of complex plots that represent many things without sufficient attempt to summarize and distill the information in an efficient manner.

There are some definitions in the first few paragraphs that are also imprecise and that need cleaning up to be correct or avoid misleading the reader. For example:

Line 1: "may" play a role in climate change. See comment below.
Also, what is "ground contamination"? This should be clarified.

Paragraph 1: "In the atmosphere bioaerosols have strong effects on the global climate, promoting cloud formation and ice nucleation events ..." The statement as written is misleading and overstated. It is known that certain types of bacteria and other microorganisms induce ice nucleation at relatively high temperatures. It is only hypothesized, however, that they directly influence cloud formation and even more these effects propagate to regional climate, much less global climate concerns. There are seeds of truth here, but the general statements are much overstated.

The highlighted sentence in the second paragraph stating "aerosols represent small water droplets containing chemical or solid particles; while bioaerosols contain microorganisms." These definitions are significantly different than what is generally accepted within the atmospheric research communities. Aerosols are typically differentiated from droplets by different sizes, in that aerosol particles form the nuclei onto which droplets form. Bioaerosols, in this context is not defined, but would typically not "contain" microorganisms, but would be represented by a microorganism, at least in one example. Also, what are "chemical particles?" This term is unclear.

The beginning of the third paragraph "Recently, we discovered that aerosols are generated by raindrops hitting the soil." The idea that rain hitting soil creates aerosols is not new, as this sentence implies. The mechanism and details are interesting and new, but the wording here is imprecise.

Again, late in the third paragraph "chemicals permeating in the soil" is really vague and should be clarified specifically.

Reviewer #4 (Remarks to the Author)

The authors have answered to all my comments and modified the manuscript by taking into account my suggestions of improvement.

More details have been included in the Material and Method section and the manuscript gained in clarity. Furthermore, statistical tests were missing and are now included in the paper. Finally, the

question of loss of culturability after aerosolization, which was my main concern, is now discussed in the text.

Then, I recommend now the paper for publication.

REVIEWERS' COMMENTS:

Reviewer #3 (Remarks to the Author):

Comment A1: The article entitled “Bioaerosol Generation by Raindrops on Soil” submitted by Joung, Ge, and Buie has been revised after taking comments from reviewers into account. After re-reading it, my feel is that the manuscript has made some improvements, especially be removing over-stated thoughts about potential application to global health. The manuscript is interesting and presents interesting experimental work.

Response A1: We appreciate the reviewer’s acknowledgement of our revised manuscript.

Comment A2: Overall, however, my feel is that the message is not crisply presented and would benefit from significant streamlining of the plots and associated message. Several figures have been chopped into multiple figures, but the total number of plots appears not to have changed. Along with this, there are so many different stories and messages presented by each figure that it is difficult to follow along in detail. There is simply not enough room in the short article to describe in text about all the messages that authors attempt to show in the figures. My feeling is that the manuscript is still too full of complex plots that represent many things without sufficient attempt to summarize and distill the information in an efficient manner.

Response A2: As the reviewer mentioned, it is challenging to explain all the details in our work with the limited length of a journal article. However, the reviewer seems to suggest a substantial revision or reorganization, which seems counter to the opinions of other reviewers. This article currently has seven sections with six figures and one table for the introduction, result, and discussion sections. We used a single figure with a detailed caption for one or two sections to try to effectively explain our findings and conclusions. In addition, we summarized each section with a topic sentence in the first line of the paragraph. The overall structure of our manuscript has (1) our previous work on aerosol generation, (2) visualization of soils with bacteria and bioaerosols, (3) viability testing of bacteria transferred by aerosols, (4) the mechanism of bioaerosol generation, (5) the effect of environmental conditions and impact conditions on aerosol generation, and (6) the aerosolization efficiency to estimate global transmission of bacteria by rain. We believe that this structure, given the space constraints, clearly presents our findings. However, if the reviewer has comments on specific sections, paragraphs, or figures that could benefit from additional text we are happy to update the manuscript accordingly.

There are some definitions in the first few paragraphs that are also imprecise and that need cleaning up to be correct or avoid misleading the reader. For example:

Comment A3: Line 1: “may” play a role in climate change. See comment below. Also, what is “ground contamination”? This should be clarified.

Response A3: As we cited the references in the manuscript, it is known that bioaerosols play an important role in climate change, human health, and agricultural productivity; however, we have added “may” in the sentence to prevent unnecessary overstatements as the reviewer originally suggested. To clarify the meaning, we have changed “water and ground contamination” to “water and soil contaminants” in the abstract.

Comment A4: Paragraph 1: “In the atmosphere bioaerosols have strong effects on the global climate, promoting cloud formation and ice nucleation events …” The statement as written is misleading and overstated. It is known that certain types of bacteria and other microorganisms induce ice nucleation at relatively high temperatures. It is only hypothesized, however, that they directly influence cloud formation and even more these effects propagate to regional climate, much less global climate concerns. There are seeds of truth here, but the general statements are much over-stated.

Response A4: We have changed the sentence to now read “In the atmosphere, bioaerosols can influence the global climate, promoting cloud formation and ice nucleation events …”

Comment A5: The highlighted sentence in the second paragraph stating “aerosols represent small water droplets containing chemical or solid particles; while bioaerosols contain microorganisms.” These definitions are significantly different than what is generally accepted within the atmospheric research communities. Aerosols are typically differentiated from droplets by different sizes, in that aerosol particles form the nuclei onto which droplets form. Bioaerosols, in this context is not defined, but would typically not “contain” microorganisms, but would be represented by a microorganism, at least in one example. Also, what are “chemical particles?” This term is unclear.

Response A5: To make the paragraph clear and to prevent misleading, we have updated the sentence as follows “In this work, aerosols represent small water droplets suspended in the air; in particular, bioaerosols are defined as aerosols containing microbes.”

Comment A6: The beginning of the third paragraph “Recently, we discovered that aerosols are generated by raindrops hitting the soil.” The idea that rain hitting soil creates aerosols is not new, as this sentence implies. The mechanism and details are interesting and new, but the wording here is imprecise.

Response A6: We have updated the sentence as following: “Recently, we discovered a new mechanism of aerosol generation by raindrops hitting soil.”

Comment A7: Again, late in the third paragraph “chemicals permeating in the soil” is really vague and should be clarified specifically.

Response A7: We have updated the sentence as following: “Furthermore, we have shown that fluorescent dyes permeated in the soil can be dispersed by aerosols.”

Reviewer #4 (Remarks to the Author):

Comment B1: The authors have answered to all my comments and modified the manuscript by taking into account my suggestions of improvement. More details have been included in the Material and Method section and the manuscript gained in clarity. Furthermore, statistical tests were missing and are now included in the paper. Finally, the question of loss of culturability after aerosolization, which was my main concern, is now discussed in the text. Then, I recommend now the paper for publication.

Response B1: We appreciate the recommendation of the reviewer for the publication of our manuscript in Nature Communications.